# PSAG: Projection-based Stabilized Attribution Guidance for Online Continual Learning

**Hang Yu**                                                                 *hayu0853@uni.sydney.edu.au*
*School of Computer Science*
*The University of Sydney*

**Kun Hu**[*]                                                                 *k.hu@ecu.edu.au*
*School of Science*
*Edith Cowan University*

**Zhuqiang Lu**                                                            *zhuqiang.lu@sydney.edu.au*
*School of Computer Science*
*The University of Sydney*

**Steven Qiang Lu**                                                     *steven.lu@sydney.edu.au*
*The University of Sydney Business School*
*The University of Sydney*

**Zhiyong Wang**                                                        *zhiyong.wang@sydney.edu.au*
*School of Computer Science*
*The University of Sydney*

**Fengxiang He**                                                          *fhe@ed.ac.uk*
*School of Informatics*
*University of Edinburgh*

**Reviewed on OpenReview:** *https://openreview.net/forum?id=NvXpSvMrXS*

## Abstract

Online Continual Learning (OCL) aims to incrementally learn from non-stationary data streams in a one-pass setting, facing the dual challenges of catastrophic forgetting and insufficient training. These challenges intensify the stability-plasticity dilemma, where preserving old knowledge conflicts with acquiring new information. In this paper, we propose Projection-based Stabilized Attribution Guidance (PSAG), a modular framework that leverages gradient-based attributions as active guidance signals to selectively preserve task-relevant representations. Our framework consists of three complementary mechanisms: (1) Attribution-Guided Feature Modulation (AGFM) that anchors critical features in the representation space; (2) Importance-Aware Loss Reweighting (IALR) that prioritizes informative samples at the loss level; and (3) Manifold-Consistent Projection (MCP) that emphasizes critical feature dimensions within a Riemannian metric space. To address the issue of attribution instability in online continual learning, we introduce a Reliable Reference Model (R-Model) that maintains consistent knowledge through exponential moving average updates. This design prevents feedback loops during attribution computation and enables reliable feature importance estimation. Extensive experiments on Split CIFAR-10, Split CIFAR-100, and Split Mini-ImageNet demonstrate that PSAG achieves consistent improvements over strong baselines, confirming the efficacy of stabilized attribution guidance in resolving the stability-plasticity dilemma.

---

[*]Corresponding author.

# 1 Introduction

Continual learning (CL) Chen & Liu (2018); Van de Ven & Tolias (2019); Wang et al. (2024); Cha & Cho (2025) enables models to accumulate knowledge incrementally across a sequence of tasks, and has gained significant attention due to its wide applicability across diverse domains Zhang et al. (2023); Dai et al. (2025). Online continual learning (OCL) Aljundi et al. (2019b); Mai et al. (2022) represents a more challenging variant where the model learns from a non-stationary data stream in a single pass. Unlike conventional CL which primarily combats catastrophic forgetting McCloskey & Cohen (1989); French (1999), OCL must also contend with limited exposure to each sample. This insufficiency amplifies parameter drift and intensifies the stability-plasticity dilemma Chaudhry et al. (2019); Prabhu et al. (2020); Sangermano et al. (2022).

Replay-based methods have emerged as a standard approach in OCL Aljundi et al. (2019a); Rolnick et al. (2019); Shim et al. (2021); Mai et al. (2021). Experience Replay (ER) Rolnick et al. (2019) mitigates forgetting by interleaving stored past samples with current data. However, most replay methods suffer from unconstrained interference between tasks. They typically assign equal weights to replayed and incoming samples, treating all input features with equal importance during optimization Ebrahimi et al. (2021). Consequently, the strong gradients generated by new tasks may inadvertently disrupt the critical feature subspaces associated with previous tasks, leading to shallow updates and severe forgetting Jung et al. (2023).

Concurrently, advances in explainable AI (XAI) have highlighted how attributions can identify input features responsible for predictions Selvaraju et al. (2017); Shrikumar et al. (2017); Sundararajan et al. (2017). Although attributions hold promise as active guidance signals, integrating them into OCL faces a critical challenge regarding instability. The current model evolves rapidly and is prone to overfitting transient batch statistics. Attributions derived directly from such a volatile model are noisy, potentially creating detrimental feedback loops if used for guidance.

To address these challenges, we propose Projection-based Stabilized Attribution Guidance (PSAG), a framework that transforms attribution from a post-hoc explanation into an active stabilizing constraint. To overcome instability, we introduce a Reliable Reference Model (R-Model), a shadow copy updated via exponential moving average, to serve as a stability anchor. Building on these stable signals, PSAG integrates three complementary mechanisms (AGFM, IALR, and MCP) to regulate the learning process at the feature, loss, and metric levels, forcing the optimization trajectory to respect the intrinsic structure of past knowledge.

The main contributions of this work are summarized as follows:

- We propose Projection-based Stabilized Attribution Guidance (PSAG), a modular framework that mitigates catastrophic forgetting by actively protecting critical features identified through stabilized attribution signals, thereby achieving an effective balance between stability and plasticity.

- We introduce the R-Model mechanism to resolve attribution instability in OCL. This design effectively prevents feedback loops during attribution computation and enables reliable feature importance estimation. We also develop a multi-level guidance system comprising Attribution-Guided Feature Modulation (AGFM) for feature anchoring, Importance-Aware Loss Reweighting (IALR) for sample prioritization, and Manifold-Consistent Projection (MCP) for metric space reshaping.

- We provide a comprehensive evaluation on Split CIFAR-10, Split CIFAR-100, and Split Mini-ImageNet. The results demonstrate that PSAG achieves consistent improvements over strong baselines and effectively reduces catastrophic forgetting.

# 2 Related Work

## 2.1 Online Continual Learning

Continual Learning Parisi et al. (2019); Klasson et al. (2023); Yu et al. (2023); Daxberger et al. (2023); Yang et al. (2024); Bhat et al. (2024); Harun & Kanan (2024); Banayeeanzade et al. (2025) and Online Continual Learning Harun et al. (2023); Huang et al. (2024); Serra et al. (2025); Saha & Roy (2025) have witnessed

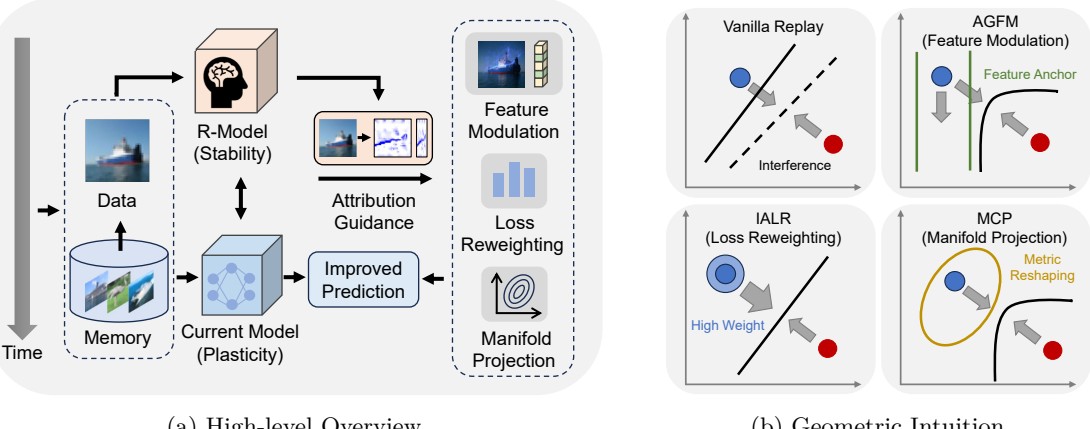

(a) High-level Overview         (b) Geometric Intuition

Figure 1: Conceptual overview of the PSAG framework. (a): A dual-branch structure where the stable R-Model guides the plastic learner via three pathways: feature modulation (AGFM), loss reweighting (IALR), and manifold projection (MCP). (b): In contrast to vanilla replay where unconstrained updates cause interference, PSAG utilizes attribution signals to anchor critical features and reshape the metric space, forcing the decision boundary to bend around protected knowledge.

significant progress. Replay-based OCL methods have been proven effective in addressing catastrophic forgetting by maintaining and revisiting a small set of past examples Chaudhry et al. (2018; 2019); Buzzega et al. (2020); Sun et al. (2022); Pham et al. (2022); Prabhu et al. (2023); Ghunaim et al. (2023). In particular, replay-based OCL methods can be broadly categorized into three main streams: memory retrieval (selecting which past samples to replay) Aljundi et al. (2019a); Shim et al. (2021), memory update (admitting new samples into the buffer) Aljundi et al. (2019b); Jin et al. (2021), and model update (utilizing replayed samples more effectively) Aljundi et al. (2019b); Gu et al. (2022); Guo et al. (2022). For instance, Chaudhry et al. (2019) proposes to store and utilize the previous samples for the current training, while Aljundi et al. (2019a) chooses the most interfered samples for the replay for jointly learning with new samples. Notably, Gu et al. (2024) utilizes contrastive learning by summarizing stream data on the prediction to preserve prior knowledge. Despite their success, current OCL methods seldom leverage insights into model decision-making to guide learning, particularly in understanding how individual decisions influence model behavior and performance Parisi et al. (2019). In this work, we investigate how to effectively utilize gradient-based attributions as active constraints in the online setting. We introduce a mechanism to stabilize these signals and project them into feature modulation and loss reweighting terms, thereby enforcing the retention of task-relevant representations against interference.

## 2.2 Explainable AI and Attribution

Explainable Artificial Intelligence (XAI) is essential for trust, transparency, and accountability, as AI systems become more complex and pervasive Gunning & Aha (2019). Yet traditional gradient-based interpretability techniques have clear shortcomings Zeiler & Fergus (2014); Smilkov et al. (2017): vanilla gradients are prone to the saturation problem Shrikumar et al. (2017), where gradients of critical features may approach zero despite their importance to the model. To address these issues, researchers have introduced attribution methods such as Integrated Gradients and DeepLIFT Sundararajan et al. (2017); Ancona et al. (2018), which focus on mining the input features most responsible for a model's prediction. XAI methods are becoming essential tools for understanding and validating AI systems across various domains Gilpin et al.. These efforts have also achieved notable success in continual learning. For example, RRR Ebrahimi et al. (2021) incorporates explanations as a regularization constraint to enforce decision consistency, while EPR Saha & Roy (2023) utilizes saliency maps to enhance the processing of memory samples. Similarly, SER Bellitto et al. (2024) employs saliency but relies on external oracles, rendering it dataset-specific and not self-contained. However, such explorations in OCL remain relatively underexplored. Our PSAG framework

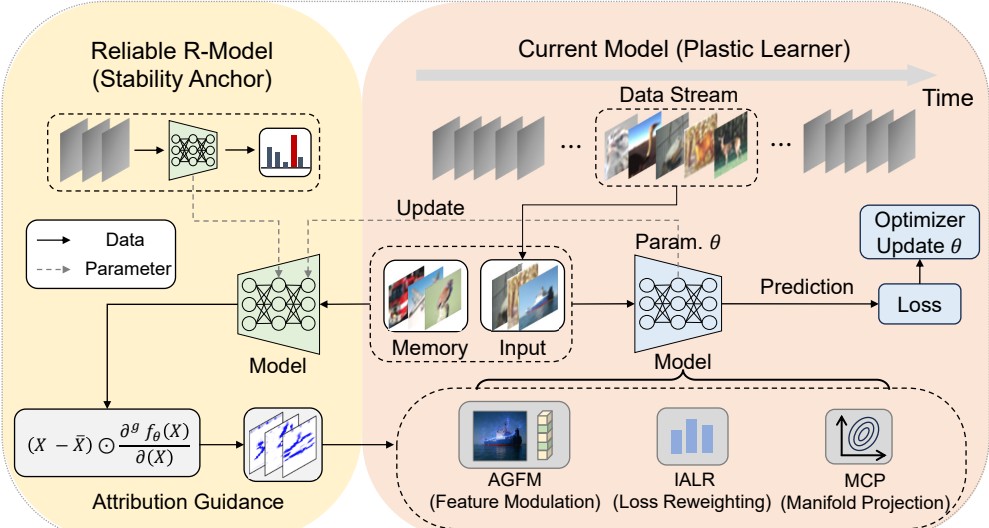

Figure 2: Detailed PSAG framework. For each mini-batch, the R-Model updated via EMA generates stable attribution maps, which are then transformed into guidance signals injected at three specific levels: (1) AGFM modulates the feature extractor output element-wise to emphasize critical channels; (2) MCP defines a Riemannian metric tensor for the contrastive objective; and (3) IALR scales the final loss based on the information energy of the samples. This self-contained loop ensures consistent knowledge preservation.

leverages stabilized attributions to explicitly anchor critical features and reweight training losses during the online update process.

## 3 Methodology

We propose Projection-based Stabilized Attribution Guidance (PSAG), a modular framework designed to resolve the stability-plasticity dilemma in OCL through gradient-based attribution methods. To intuitively understand the mechanism, Figure 1 visualizes the decision boundary dynamics in a simplified feature space. Unlike vanilla replay where destructive interference from new tasks inadvertently pushes decision boundaries into previous knowledge regions, PSAG counters this through three complementary geometric constraints: AGFM acts as a feature barrier to anchor critical dimensions, IALR amplifies the gradient resistance of informative samples, and MCP reshapes the metric space into a Riemannian manifold. Together, these mechanisms force the optimization trajectory to respect the intrinsic geometry of past knowledge, thereby minimizing catastrophic forgetting.

The detailed architecture implementing these concepts is depicted in Figure 2. The training process involves an interplay between the plastic current model and a Reliable Knowledge Reservoir (R-Model). Specifically, inputs are first passed through the frozen R-Model to compute spatial attribution maps. A specialized projector then aligns these spatial signals with the feature channel dimensions. These signals are subsequently injected into the learning process through three distinct mechanisms: AGFM performs element-wise modulation on intermediate features, MCP provides a metric matrix for the contrastive loss calculation, and IALR adjusts the scalar weight of the total loss. The R-Model is periodically updated via an exponential moving average of the current model's parameters, closing the loop for stabilized self-evolution.

### 3.1 Problem Definition

In the online continual learning (OCL) setting, a model $f_\theta$ with parameters $\theta$ learns from a non-stationary data stream segmented into tasks $\mathcal{T}_1, \mathcal{T}_2, \ldots, \mathcal{T}_N$. During task $\mathcal{T}_i$, the model observes a single-pass sequence

of labeled samples $(X_{i,j}, Y_{i,j})$. The learning objective is to update $\theta$ to fit the current task while retaining knowledge from previous tasks.

Following the replay-based approach Rolnick et al. (2019), we maintain a memory buffer $\mathcal{M}$ of capacity $|\mathcal{M}|$ storing samples from completed tasks. During training on $\mathcal{T}_i$, each optimization step draws a mini-batch $B_i \subset \mathcal{T}_i$ from the current task and $B_{\mathcal{M}} \subset \mathcal{M}$ from memory. The combined batch $B = B_i \cup B_{\mathcal{M}}$ is used to update the model, balancing plasticity (learning new knowledge) and stability (preserving old knowledge).

## 3.2 Attribution for Feature Importance

Attribution methods quantify each input feature's contribution to a model's prediction. Given a model $f_\theta$, input $X$, and predicted label $\hat{Y} = f_\theta(X)$, an attribution map $A(X, \hat{Y}, f_\theta)$ assigns importance scores to input dimensions.

We employ gradient-based attribution Shrikumar et al. (2016), computing:

$$A(X, \hat{Y}, f_\theta) = X \odot \nabla_X f_\theta(X), \tag{1}$$

where $\odot$ denotes element-wise multiplication. For computational efficiency in the online setting, we use a discrete approximation Shrikumar et al. (2017):

$$A(X, \hat{Y}, f_\theta) = \Delta X \odot \frac{\partial^g f_\theta(X)}{\partial X}, \tag{2}$$

where $\Delta X = X - \bar{X}$ is the difference from a reference input $\bar{X}$ (e.g., gray baseline), and $\frac{\partial^g f_\theta(X)}{\partial X}$ approximates the gradient through finite differences. This requires only a single forward-backward pass while maintaining attribution quality. Specifically, the term $\frac{\partial f_\theta(X)}{\partial X}$ in our formulation serves as a continuous proxy for the discrete slope defined in DeepLIFT as $m_{\Delta x} = \frac{f(x) - f(\bar{x})}{x - \bar{x}}$. By using this finite difference formulation, we attribute the difference in output $\Delta y$ to the difference in input $\Delta x$ relative to a baseline $\bar{x}$, thereby avoiding the gradient saturation problem common in standard sensitivity maps.

## 3.3 Reliable Reference Model

In OCL, the model parameters $\theta$ evolve continuously, causing attribution maps to drift over time. To provide stable attribution signals, we maintain a reliable reference model (R-Model) $f_{\theta_{\text{ref}}}$ with the same architecture but slower weight updates.

The R-Model is initialized after the first task and updated periodically using exponential moving average (EMA):

$$\theta_{\text{ref}} \leftarrow (1 - \eta) \cdot \theta_{\text{ref}} + \eta \cdot \theta, \tag{3}$$

where $\eta \in [0, 1)$ is the EMA update rate, applied every $K$ optimization steps. The momentum coefficient controls the update speed, and we employ a scheduled $\eta$ to balance initial adaptation and long-term stability. This slow-update schedule allows the R-Model to absorb new knowledge gradually while maintaining consistency, serving as a stable reference for computing attributions throughout the learning process.

## 3.4 Projection-based Stabilized Attribution Guidance

We propose three complementary mechanisms that leverage XAI attributions to guide online continual learning at different levels: feature modulation, loss reweighting, and metric projection.

**Attribution-Guided Feature Modulation (AGFM)** Contrastive learning effectively maintains discriminative representations Cha et al. (2021); Mai et al. (2021). AGFM enhances this by anchoring critical patterns identified by the R-Model. Specifically, we define a feature-level importance vector $\mathbf{v} \in \mathbb{R}^C$ by aggregating the spatial attribution map $A \in \mathbb{R}^{C \times H \times W}$ via global average pooling:

$$\mathbf{v}_c = \frac{1}{H \times W} \sum_{h=1}^{H} \sum_{w=1}^{W} |A(c, h, w)|. \tag{4}$$

Let $\mathcal{N}(\cdot)$ denote the min-max normalization operator defined as $\mathcal{N}(\mathbf{x}) = \frac{\mathbf{x}-\min(\mathbf{x})}{\max(\mathbf{x})-\min(\mathbf{x})+\epsilon}$, where $\epsilon$ is a small constant for numerical stability. We compute the modulation coefficient:

$$\mathbf{m} = \mathbf{1} + \lambda_{\text{AGFM}} \cdot \mathcal{N}(\mathbf{v}). \tag{5}$$

The feature representation $z$ is then modulated element-wise as $\tilde{z} = z \odot \mathbf{m}$, where $\odot$ denotes the Hadamard product. This modulation forces the contrastive objective to prioritize feature channels that are causally significant for the R-Model's predictions.

**Importance-Aware Loss Reweighting (IALR)**  While AGFM operates on internal representations, IALR provides a generalized loss-level prioritization. We hypothesize that samples with higher total attribution energy contain denser semantic information crucial for memory retention.

For a batch $B$, we quantify the information density of the $i$-th sample as $E_i = \|A_i\|_1$. We apply the same normalization operator $\mathcal{N}$ (computed over the batch dimension) to derive the reweighting scalar $w_i$:

$$w_i = 1 + \alpha_{\text{IALR}} \cdot (\mathcal{N}(E)_i - 0.5), \tag{6}$$

where $\mathcal{N}(E)_i$ is the normalized energy score of sample $i$ within the current batch. This mechanism focuses gradient updates on samples with high epistemic value.

**Manifold-Consistent Projection (MCP)**  Standard contrastive learning assumes a Euclidean metric space, which treats all feature dimensions isotropically. To enforce geometric stability, we propose learning on a Riemannian manifold induced by the attribution field.

We define a local metric tensor $G \in \mathbb{R}^{D \times D}$ based on the spatial importance distribution. For computational feasibility, we approximate $G$ as a diagonal matrix derived from the flattened spatial attribution:

$$G = \text{diag}\left(\text{vec}\left(\sum_c |A_c|\right)\right) + \epsilon I. \tag{7}$$

The distance between two representations $z_i$ and $z_j$ is then measured using the Mahalanobis distance induced by $G$:

$$d_{\mathcal{M}}(z_i, z_j) = \sqrt{(z_i - z_j)^\top G (z_i - z_j)}. \tag{8}$$

By substituting the Euclidean distance with $d_{\mathcal{M}}$ in the contrastive loss, MCP penalizes feature distortions specifically along the manifold directions identified as critical by the R-Model.

### 3.5 Overall Framework

The proposed mechanisms are integrated into a unified training objective. The final loss function for a mini-batch is a weighted combination of the cross-entropy loss and the modulated contrastive loss:

$$\mathcal{L}_{\text{total}} = \frac{1}{|B|} \sum_{i \in B} w_i \cdot (\mathcal{L}_{\text{CE}}(\hat{y}_i, y_i) + \lambda_{\text{SCL}} \cdot \mathcal{L}_{\text{SCL}}(\tilde{z}_i, Y_B; d_{\mathcal{M}})), \tag{9}$$

where $w_i$ is provided by IALR, $\tilde{z}_i$ is the AGFM-modulated feature, and $d_{\mathcal{M}}$ denotes that the contrastive loss utilizes the MCP metric. This formulation ensures that the model optimization is consistently guided by the stable attribution signals from the R-Model across feature, metric, and loss levels.

## 4 Experiments

We evaluate the effectiveness of PSAG on three standard OCL benchmarks. We first detail the experimental setup, followed by a comparison with strong baselines. Finally, we provide comprehensive ablation studies and sensitivity analyses to validate the contribution of each proposed mechanism.

---

**Algorithm 1** Projection-based Stabilized Attribution Guidance (PSAG) for OCL

---

**Require:** Model $f_\theta$, data stream $S$, memory $\mathcal{M}$, hyperparameters $\lambda, \alpha, \eta$
 1: Initialize R-Model $\theta_{\text{ref}} \leftarrow \theta$ after learning Task 1
 2: **for** Task $\mathcal{T}_i$ in $\mathcal{S}$ **do**
 3:     **for** Mini-batch $(X_i, Y_i)$ in $\mathcal{T}_i$ **do**
 4:         $(X_\mathcal{M}, Y_\mathcal{M}) \leftarrow \text{Sample}(\mathcal{M})$
 5:         $X \leftarrow [X_i; X_\mathcal{M}], \quad Y \leftarrow [Y_i; Y_\mathcal{M}]$
 6:         $\mathcal{A} \leftarrow A(X, Y, f_{\theta_{ref}})$               ▷ Compute stable attributions
 7:         $\mathbf{E} \leftarrow \|\mathcal{A}\|_1$               ▷ Information Energy for IALR
 8:         $\mathbf{V} \leftarrow \text{SpatialMean}(|\mathcal{A}|)$        ▷ Channel Importance for AGFM
 9:         $G \leftarrow \text{diag}(\text{vec}(\sum \mathcal{A}))$         ▷ Metric Tensor for MCP
10:         $Z \leftarrow \phi_\theta(X)$                 ▷ Extract features
11:         $\tilde{Z} \leftarrow Z \odot (\mathbf{1} + \lambda \cdot \mathcal{N}(\mathbf{V}))$    ▷ Feature modulation Eq. equation 5
12:         $\mathcal{L}_{\text{CE}} \leftarrow \text{CrossEntropy}(h_\theta(Z), Y)$
13:         $\mathcal{L}_{\text{SCL}} \leftarrow \text{ContrastiveLoss}(\tilde{Z}, Y; d_\mathcal{M})$    ▷ Use $d_\mathcal{M}$ from Eq. equation 8
14:         $\mathbf{w} \leftarrow 1 + \alpha \cdot (\mathcal{N}(\mathbf{E}) - 0.5)$    ▷ Sample weights Eq. equation 6
15:         $\mathcal{L}_{total} \leftarrow \frac{1}{|B|} \sum (\mathbf{w} \odot (\mathcal{L}_{\text{CE}} + \mathcal{L}_{\text{SCL}}))$
16:         Update $\theta \leftarrow \theta - \eta_{lr} \nabla_\theta \mathcal{L}_{total}$
17:         Update $\mathcal{M}$ with $(X_i, Y_i)$
18:         **if** $it \bmod K = 0$ **then**
19:             $\theta_{ref} \leftarrow (1 - \eta) \cdot \theta_{ref} + \eta \cdot \theta$
20:         **end if**
21:         $it \leftarrow it + 1$
22:     **end for**
23: **end for**

---

## 4.1 Experimental Setup

**Datasets.** We verify our method on three widely used OCL benchmarks: **Split CIFAR-10** Krizhevsky et al. (2009), **Split CIFAR-100** Krizhevsky et al. (2009), and **Split Mini-ImageNet** Vinyals et al. (2016). CIFAR-10 and CIFAR-100 consist of 50,000 training and 10,000 test images, categorized into 10 and 100 classes, respectively. Mini-ImageNet contains 100 classes with 600 images per class. Following standard protocols Shim et al. (2021), Split CIFAR-10 is divided into 5 tasks (2 classes per task), while Split CIFAR-100 and Split Mini-ImageNet are divided into 10 tasks (10 classes per task).

**Baselines & Metrics.** We compare PSAG against representative OCL methods, including standard, proxy-based, and contrastive-based experience replay baselines (ER Rolnick et al. (2019), MIR Aljundi et al. (2019a), ASER Shim et al. (2021), SSIL Ahn et al. (2021), LAS Huang et al. (2024), SCR Mai et al. (2021) and SSCR Gu et al. (2024)). We report the **Average Accuracy (ACC)**, defined as the mean test accuracy across all tasks after training concludes:

$$A_T = \frac{1}{T} \sum_{t=1}^{T} a_{T,t},$$

where $a_{T,t}$ is the accuracy on task $t$ after learning the final task $T$.

**Implementation Details.** Consistent with prior works Shim et al. (2021); Gu et al. (2024), we use a Reduced ResNet18 as the backbone. All methods are trained from scratch using an SGD optimizer with a learning rate of 0.1. The memory buffer sizes are set identical to the compared methods to ensure fair comparison. All experiments are conducted on a single NVIDIA GeForce RTX 4090 GPU. Detailed hyperparameter configurations are provided in Appendix A.2.

Table 1: Performance comparison on CIFAR-10, CIFAR-100, and Mini-ImageNet across different memory buffer sizes ($M$). Results are reported as mean $\pm$ 95% confidence interval over 5 runs.

| Methods | CIFAR-10 | | | CIFAR-100 | | | Mini-ImageNet | | |
|---|---|---|---|---|---|---|---|---|---|
| | $M = 200$ | $M = 500$ | $M = 1000$ | $M = 1000$ | $M = 2000$ | $M = 5000$ | $M = 1000$ | $M = 2000$ | $M = 5000$ |
| ER | 21.41$\pm$1.11 | 24.38$\pm$2.17 | 28.12$\pm$3.21 | 8.88$\pm$0.53 | 10.71$\pm$0.70 | 16.06$\pm$1.00 | 8.59$\pm$0.89 | 9.65$\pm$1.44 | 16.98$\pm$2.08 |
| MIR | 21.60$\pm$0.88 | 24.58$\pm$2.21 | 30.13$\pm$4.48 | 9.20$\pm$0.48 | 10.59$\pm$0.96 | 15.66$\pm$1.26 | 8.80$\pm$0.45 | 10.04$\pm$1.33 | 17.24$\pm$1.71 |
| ASER | 24.57$\pm$6.71 | 29.70$\pm$4.13 | 35.15$\pm$1.65 | 13.36$\pm$1.18 | 16.56$\pm$0.67 | 22.12$\pm$0.73 | 15.18$\pm$1.85 | 18.21$\pm$0.73 | 24.23$\pm$1.44 |
| SSIL | 32.64$\pm$4.29 | 35.00$\pm$2.08 | 37.86$\pm$3.43 | 14.06$\pm$1.38 | 16.04$\pm$1.44 | 20.07$\pm$1.57 | 14.63$\pm$1.44 | 15.86$\pm$1.49 | 21.81$\pm$2.29 |
| ACE | 36.37$\pm$2.33 | 40.63$\pm$3.32 | 43.40$\pm$3.67 | 13.62$\pm$0.81 | 15.21$\pm$1.46 | 20.37$\pm$0.68 | 12.44$\pm$2.57 | 15.58$\pm$0.90 | 20.43$\pm$2.10 |
| LAS | 23.50$\pm$2.99 | 33.24$\pm$4.86 | 42.45$\pm$3.98 | 14.12$\pm$1.26 | 18.96$\pm$0.57 | 25.78$\pm$2.66 | 15.67$\pm$0.83 | 18.35$\pm$1.38 | 23.50$\pm$3.24 |
| SCR | 44.40$\pm$3.55 | 55.06$\pm$2.04 | 61.11$\pm$1.53 | 21.14$\pm$0.81 | 24.67$\pm$0.98 | 27.48$\pm$0.58 | 19.99$\pm$0.61 | 23.26$\pm$0.75 | 25.49$\pm$1.10 |
| SSCR | 44.10$\pm$2.23 | 55.98$\pm$1.15 | 61.73$\pm$0.68 | 22.84$\pm$0.42 | 25.25$\pm$0.70 | 26.98$\pm$1.31 | 20.36$\pm$0.62 | 23.26$\pm$1.03 | 25.81$\pm$0.70 |
| **PSAG** | **45.72$\pm$2.16** | **57.39$\pm$2.22** | **62.44$\pm$1.06** | **23.46$\pm$0.66** | **26.32$\pm$1.37** | **28.42$\pm$0.29** | **21.09$\pm$0.48** | **24.07$\pm$1.00** | **26.61$\pm$1.30** |

## 4.2 Overall Performance

**Main Results.** Table 1 presents the performance comparison of our proposed PSAG against state-of-the-art baselines across three benchmark datasets. For all main results reported in Table 1, we employ the DeepLIFT formulation (Eq. 2) using a gray image baseline ($\bar{X} = 0.5$) to compute attribution maps. PSAG consistently outperforms all baselines, demonstrating the effectiveness of stabilized attribution guidance in mitigating catastrophic forgetting.

**Mitigating catastrophic interference.** Traditional replay methods (e.g., ER, MIR) treat all features equally, causing new task gradients to blindly overwrite critical patterns of old tasks. PSAG addresses this through attribution-guided stabilization: the R-Model (Reliable Knowledge Reservoir) provides stable signals to identify "what matters," enabling selective protection of critical knowledge while maintaining plasticity.

**Consistent improvements over strong baselines.** PSAG achieves substantial gains over the strongest baseline SSCR: +1.41% on CIFAR-10 (M=500), +1.44% on CIFAR-100 (M=5000), and +0.81% on Mini-ImageNet (M=2000). These gains are particularly notable given that SSCR already incorporates sophisticated coreset selection, demonstrating that stabilized attribution guidance provides benefits orthogonal to memory selection strategies.

**Robustness across memory budgets.** The performance advantage persists across diverse memory regimes, from limited budgets (M=200 on CIFAR-10) to larger capacities (M=5000 on CIFAR-100). This validates the scalability of our approach: PSAG enhances representation learning regardless of the replay buffer size.

**Task-wise Performance Analysis.** Figure 3 presents per-task accuracy measured after completing all tasks, providing a fine-grained view of catastrophic forgetting, the core challenge in continual learning.

**Consistent task-level improvements.** PSAG outperforms SSCR on 4 out of 5 tasks on CIFAR-10 (Tasks 1, 3, 4, 5) and 6 out of 10 tasks on CIFAR-100. This broad coverage demonstrates that the R-Model's attribution signals successfully identify and protect critical knowledge across diverse tasks. The average improvements of +1.41% (CIFAR-10) and +1.44% (CIFAR-100) validate the effectiveness of stabilized guidance in mitigating forgetting.

**Selective strengthening on vulnerable tasks.** PSAG achieves particularly large gains on certain tasks: +3.10% on CIFAR-10 Task 1 and +7.92% on CIFAR-100 Task 7. These tasks likely suffer more severe interference in vanilla replay due to feature distribution shifts or inter-task competition. PSAG's attribution-guided mechanisms (AGFM, IALR, MCP) selectively reinforce these vulnerable tasks by identifying their critical features and preventing blind gradient updates from corrupting them.

**Stable overall trend.** While some tasks show modest or slightly negative changes (e.g., CIFAR-10 Task 2: -0.45%), the overall trend is strongly positive, and the average improvement is statistically significant. This demonstrates that PSAG provides robust benefits without catastrophically destabilizing any specific task, a key requirement for practical continual learning systems.

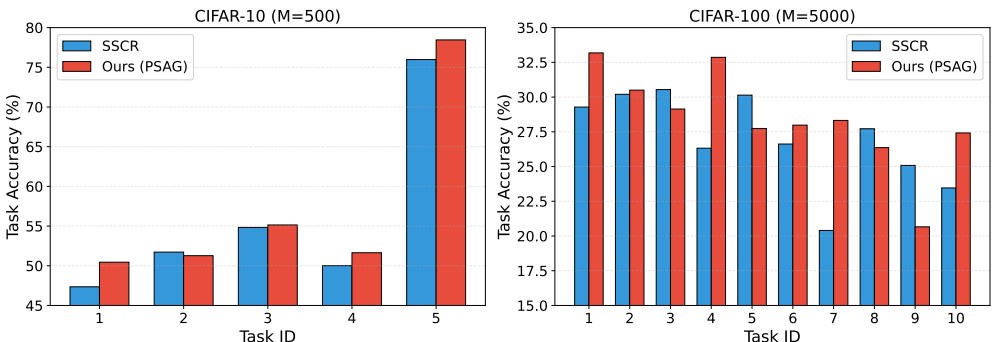

Figure 3: Task-wise accuracy comparison on CIFAR-10 ($M = 500$) and CIFAR-100 ($M = 5000$). Bars represent accuracy on specific tasks after the full training sequence. PSAG shows consistent advantages across most tasks compared to SSCR, demonstrating effective preservation of old knowledge.

**Implications for attribution guidance.** The task-wise breakdown confirms that stabilized attribution (via the R-Model) is more reliable than naively using the current model's attributions, which can be corrupted by overfitting to recent data. The R-Model's EMA-smoothed parameters filter out high-frequency noise, preserving long-term knowledge signals that guide effective stabilization.

### 4.3 Ablation Study

To dissect the individual and combined effects of PSAG's three mechanisms, we conduct ablation studies on three datasets using SSCR as the base method. Table 2 presents the results.

**Individual Component Analysis.** Each mechanism addresses a different aspect of the stability-plasticity dilemma, while its empirical effect varies across datasets. AGFM provides feature-level anchoring by amplifying responses in critical channels, and it is particularly effective on CIFAR-10. IALR performs loss-level prioritization based on attribution energy, encouraging the model to focus on high-information samples. MCP reshapes the contrastive metric space according to attribution-induced importance and shows stronger effects on more complex datasets such as CIFAR-100 and Mini-ImageNet.

**Dataset-Dependent Complementarity.** The ablation results show that the three mechanisms are complementary but not uniformly additive across datasets. On CIFAR-10, attribution-guided feature and loss guidance already capture most of the improvement, while adding MCP provides limited additional benefit under the strict full configuration. On CIFAR-100, the full configuration achieves the strongest performance, suggesting that feature-, loss-, and metric-level guidance can jointly improve stability on more complex class-incremental streams. On Mini-ImageNet, MCP-related configurations are competitive, indicating that metric-level guidance is useful when the representation manifold is more complex. Overall, PSAG should be interpreted as a modular framework whose best-performing component combination may vary with dataset complexity.

### 4.4 Sensitivity Analysis

We analyze PSAG's sensitivity to key hyperparameters controlling the strength of attribution guidance. Table 3 summarizes the impact of AGFM modulation weight ($w$) and IALR coefficient ($\alpha$) on CIFAR-10 ($M = 500$).

**Guidance Strength ($w$ and $\alpha$).** Performance improves as the modulation weight $w$ increases, peaking at $w = 2.0$ (57.39%). This confirms that stronger anchoring of critical channels is beneficial, though excessive modulation ($w > 2.5$) may overly rigidify the feature space. Similarly, the loss weighting coefficient $\alpha$ shows optimal performance at $\alpha = 2.0$, effectively prioritizing high-attribution samples without overfitting. Both parameters exhibit smooth performance curves, indicating robustness to hyperparameter choices.

Table 2: Ablation study dissecting the contribution of each component in PSAG. **AGFM**: Feature Modulation, **IALR**: Loss Reweighting, **MCP**: Manifold Projection. Experiments conducted on CIFAR-10 ($M = 500$), CIFAR-100 ($M = 5000$), and Mini-ImageNet ($M = 5000$). Results averaged over 5 runs with 95% confidence intervals.

| Configuration | Components | | | Accuracy (%) | | |
|---|---|---|---|---|---|---|
| | **AGFM** | **IALR** | **MCP** | **CIFAR-10** | **CIFAR-100** | **Mini-ImageNet** |
| SSCR (Base) | | | | 55.98±1.15 | 26.98±1.31 | 25.81±0.70 |
| + AGFM only | ✓ | | | 57.39±2.22 | 27.56±0.47 | 26.08±0.99 |
| + IALR only | | ✓ | | 57.29±3.13 | 27.53±1.37 | 25.91±0.87 |
| + MCP only | | | ✓ | 54.90±1.14 | 28.26±0.92 | 26.59±1.60 |
| + AGFM & IALR | ✓ | ✓ | | 56.52±1.22 | 27.44±0.82 | 25.67±0.74 |
| + AGFM & MCP | ✓ | | ✓ | 56.44±2.05 | 27.78±0.77 | 26.61±1.30 |
| + IALR & MCP | | ✓ | ✓ | 57.20±1.16 | 27.76±0.65 | 25.67±0.74 |
| + AGFM & IALR & MCP | ✓ | ✓ | ✓ | 56.86±4.04 | 28.50±0.37 | 26.60±0.93 |

Table 3: Sensitivity analysis of key hyperparameters on CIFAR-10 ($M = 500$). $w$: AGFM feature modulation strength, $\alpha$: IALR loss weighting coefficient. Results averaged over 5 runs with 95% confidence intervals.

| AGFM Weight ($w$) | | Accuracy (%) | IALR Weight ($\alpha$) | | Accuracy (%) |
|---|---|---|---|---|---|
| **Value** | **Note** | | **Value** | **Note** | |
| 0.0 | Baseline | 55.98±1.15 | 0.0 | Baseline | 55.98±1.15 |
| 0.5 | | 56.82±6.82 | 0.1 | | 55.32±2.63 |
| 1.0 | | 57.07±1.84 | 0.5 | | 56.41±2.37 |
| 1.5 | | 56.80±5.27 | 1.0 | | 56.89±2.77 |
| **2.0** | **Optimal** | **57.39±2.22** | **2.0** | **Optimal** | **57.29±3.13** |
| 3.0 | | 56.97±1.20 | 5.0 | | 57.05±5.00 |

**R-Model Stability Parameters.** We further investigated the update schedule of the R-Model (detailed in Appendix A.2.3). We use the update rate of $\eta_{init} = 0.1$ for the early phase, $\eta_{later} = 0.001$ for the stable phase and medium update frequency ($K = 50$) to balance plasticity and stability. This validates that the R-Model must evolve slowly to filter high-frequency noise while remaining relevant to the current task.

**Statistical Analysis.** Given the sensitivity of $p$-values to small sample sizes ($N = 5$), we analyzed effect sizes to confirm robustness. PSAG achieves a large effect ($d > 0.8$) in 8 out of 9 settings (e.g., $d = 1.88$ on CIFAR-100, $M = 5000$), indicating substantial improvements where details are shown in Appendix A.2.7.

**Comparison with EMA-only distillation.** To test whether the gain comes merely from using an EMA teacher, we add an EMA-only distillation baseline on Split CIFAR-10 with $M = 500$. It follows the same EMA update schedule as the R-Model but disables AGFM, IALR, and MCP. As shown in Table 4, EMA-only distillation improves over SSCR, confirming that EMA stabilization is helpful. However, PSAG still achieves higher accuracy and lower forgetting, suggesting that stabilized attribution guidance provides additional benefits beyond generic EMA distillation.

## 4.5 Mechanism Verification

While our main results demonstrate the superiority of PSAG when combined with strong contrastive baselines, the aggressive representation shaping of contrastive learning can sometimes mask the subtle geometric corrections introduced by attribution guidance. To explicitly visualize how PSAG mitigates interference and preserves feature structures, we conduct a fine-grained analysis using the vanilla Experience Replay (ER) baseline. In this controlled setting, denoted as PSAG-ER (detailed in Appendix A.2.5), we apply attribution-guided modulation to isolate the stabilization effect.

Table 4: EMA-only fairness comparison on Split CIFAR-10 with $M = 500$. EMA-only distillation uses the same EMA teacher schedule as PSAG but removes attribution-guided modules. Results are averaged over 5 runs with 95% confidence intervals.

| Method | Attribution | EMA Teacher | Accuracy (%) | Forgetting (%) |
|---|---|---|---|---|
| SSCR | No | No | 55.98±1.43 | 27.20±2.59 |
| EMA-only distillation | No | Yes | 57.03±2.86 | 26.81±2.73 |
| PSAG | Yes | Yes | 57.39±2.76 | 25.38±2.98 |

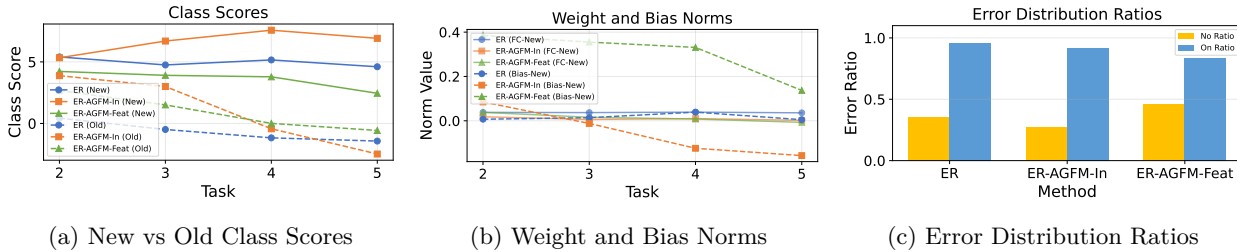

(a) New vs Old Class Scores      (b) Weight and Bias Norms      (c) Error Distribution Ratios

Figure 4: Mechanism verification comparing standard ER with PSAG-ER on CIFAR-10. PSAG-ER maintains higher class confidence and balanced weight norms, reducing inter-task confusion.

**Error Analysis.** Figure 4 presents a detailed error analysis comparing standard ER with PSAG-ER. The first panel (Class Scores) tracks the model's confidence. While standard ER exhibits declining confidence for past tasks (catastrophic forgetting) and struggles to confidently classify new data (insufficient plasticity), PSAG-ER maintains consistently higher class scores across both past and current tasks. This indicates that attribution guidance effectively enhances the model's ability to balance stability and plasticity. The second panel (Weight Norms) reveals that standard ER tends to develop a bias toward new tasks (higher norms for current task weights). In contrast, PSAG-ER balances the weight norms between past and current labels, suggesting that the model remains responsive to memory samples throughout training. The final panel (Error Ratios) decomposes misclassifications. PSAG-ER effectively reduces "No Ratio" errors (misclassifying current samples as past classes) while maintaining low "On Ratio" errors, indicating more precise decision boundaries that respect both old and new knowledge.

**Norm Perspective.** We analyze attribution maps through their norms to understand feature importance dynamics. As shown in Figure 5, there is a significant disparity in attribution patterns: memory samples consistently exhibit higher attribution norms compared to current samples. This suggests that memory samples, having been selected and replayed, carry denser semantic information crucial for the model's decision-making. PSAG exploits this by using the R-Model to explicitly anchor these high-norm features, preventing them from being eroded by new task updates.

**Attribution-Performance Correlation.** Figure 6 further quantifies the relationship between attribution strength and model performance. We observe a positive correlation trend (e.g., Pearson coefficient 0.58 for Frobenius norm) between the attribution norms of memory samples and the final OCL accuracy. Given the limited sample size ($N = 4$), we interpret this correlation as suggestive rather than statistically significant. Nevertheless, this finding provides an empirical motivation for our framework: stronger, well-preserved attribution signals appear to be associated with better knowledge retention. By enforcing consistency with the R-Model's attributions, PSAG explicitly aims to optimize this property.

**Visualizing Representational Drift.** Figure 7 visualizes the evolution of attribution maps for Task 1 samples during the standard ER training process. We observe a clear degradation in attribution quality: initially, the model focuses on semantically meaningful object shapes. However, as training progresses to later tasks, these maps become scattered and noisy, shifting towards texture-biased patterns, a symptom of overfitting to new data and forgetting old concepts. This phenomenon empirically motivates the design of

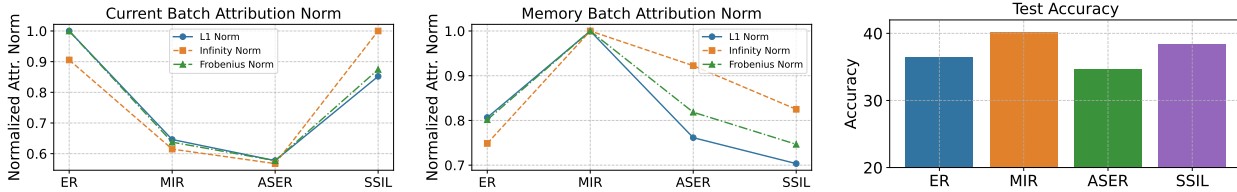

Figure 5: Evolution of attribution norms and test accuracy. Memory samples (center) maintain higher attribution energy, which correlates with better performance.

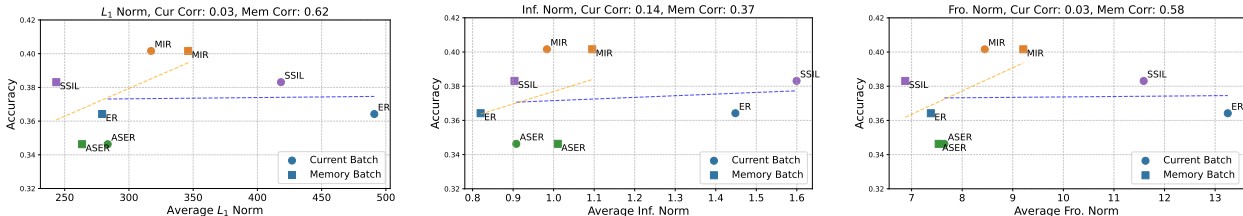

Figure 6: Correlation analysis. The strong linear relationship between memory attribution norms and accuracy validates our motivation to use attributions as guidance signals.

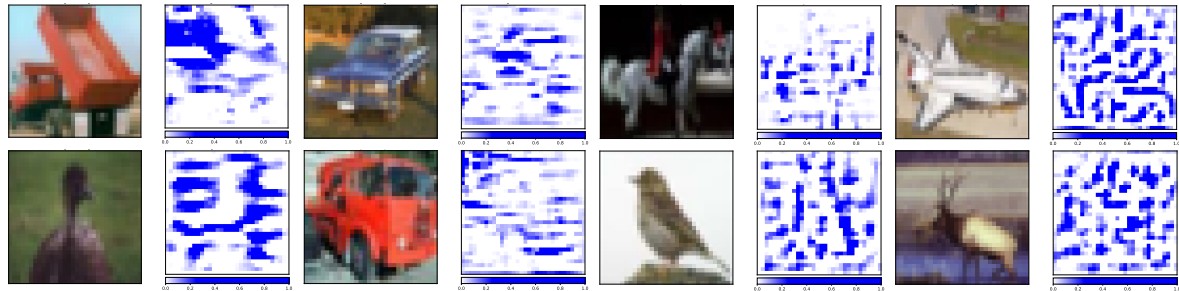

Figure 7: Visualization of attribution evolution in ER. As the model learns new tasks (left to right) with the current and memory batch (top to bottom), its attention on early task samples degrades from shapes to scattered noise. This representational drift underscores the necessity of the R-Model as a stability anchor.

Table 5: Training time comparison on CIFAR-10 ($M = 500$). Reported times are averaged over 5 runs.

| Method | ER | MIR | ASER | SSIL | ACE | LAS | SCR | SSCR | PSAG |
|---|---|---|---|---|---|---|---|---|---|
| **Time (s)** | 30.4 | 60.0 | 60.5 | 35.5 | 31.1 | 26.3 | 68.6 | 104.4 | 117.4 |
| **Acc (%)** | 24.38 | 24.58 | 29.70 | 35.00 | 40.63 | 33.24 | 55.06 | 55.98 | 57.39 |

PSAG: since the plastic learner's own attributions are unstable over time, a separate R-Model is essential to preserve and distill these focused attention patterns back into the learning process.

**Computational Complexity.** Finally, we evaluate the training-time overhead introduced by PSAG. As shown in Table 5, PSAG incurs higher cost than simple replay methods but remains close to the strongest baseline SSCR. Compared with SSCR, PSAG adds about 13 seconds of training time on CIFAR-10 ($M = 500$), corresponding to a moderate overhead of roughly 12%, while improving accuracy by 1.41 percentage points. Detailed runtime profiling is provided in Appendix A.2.11.

## 5 Conclusion

In this work, we introduced Projection-based Stabilized Attribution Guidance (PSAG), a framework that uses stabilized attribution maps as active constraints for online continual learning. By maintaining a Reliable Reference Model (R-Model), PSAG provides stable attribution signals and integrates them through three mechanisms: AGFM for feature anchoring, IALR for sample prioritization, and MCP for metric-space reshaping. Experiments on Split CIFAR-10, Split CIFAR-100, and Split Mini-ImageNet show that PSAG improves over strong replay-based baselines and reduces catastrophic forgetting across different memory budgets. Additional ablations indicate that the component effects are dataset-dependent and that attribution guidance provides benefits beyond generic EMA distillation. These results suggest that attribution signals can serve not only as post-hoc explanations, but also as practical guidance for improving stability in online continual learning.

## Broader Impact Statement

Our research contributes to the development of more reliable and interpretable adaptive AI systems. By improving stability in online continual learning, PSAG may reduce the need for repeated retraining in non-stationary environments. However, stability mechanisms may also risk anchoring early biases, making them harder to correct with new data. Practitioners should weigh the benefits of stabilized learning against the additional cost of attribution computation, especially for deployment on resource-limited devices.

## Data and code availability

The PSAG implementation and main experimental scripts are available at `https://github.com/jarrettyu/PSAG`. The datasets used in this work are publicly available benchmarks.

## Acknowledgments

H.Y. was supported in part by the JD Technology Research Scholarship in Artificial Intelligence. K.H. acknowledges funding support from the ECU Early-Mid Career Researcher (EMCR) Grant Scheme.

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

# A  Appendix

## A.1  Theoretical Analysis of Gradient Modulation

In this section, we analyze how the proposed Attribution-Guided Feature Modulation (AGFM) guides the process of Online Continual Learning (OCL) from a gradient perspective. We consider two levels of modulation: Input-level AGFM (operating on raw pixels, corresponding to the PSAG-ER variant) and Feature-level AGFM (operating on latent representations, consistent with the main PSAG framework).

**Preliminaries.**  We consider the loss function $\mathcal{L}$ for a single sample $(X, Y)$. The current and reference model parameters are $\theta$ and $\theta_{ref}$.

**Input-level AGFM:** The input $X$ is transformed into $X_{mod} = X \odot M_{in}$, where the mask $M_{in} = (1 + \beta_{in} \cdot A_{norm})$.

**Feature-level AGFM:** The representation map $F$ is modulated to $F_{mod} = F \odot M_{feat}$, where $M_{feat} = (1 + \beta_{feat} \cdot A_{feat})$. Note that during gradient computation for $\theta$, the attribution maps and masks are treated as fixed constants derived from the R-Model.

**Gradient Analysis for Input-level AGFM**  We analyze the impact on the gradient w.r.t. the original input $X$. Let $X_j$ be the $j$-th element of $X$, and $X_{mod,p}$ be the $p$-th element of $X_{mod}$. The chain rule gives:

$$\frac{\partial \mathcal{L}}{\partial X_j} = \sum_p \frac{\partial \mathcal{L}}{\partial X_{mod,p}} \frac{\partial X_{mod,p}}{\partial X_j}. \tag{10}$$

Since $X_{mod,p} = X_p \cdot M_{in,p}$ (element-wise) and $M_{in}$ is constant:

$$\frac{\partial \mathcal{L}}{\partial X_j} = \frac{\partial \mathcal{L}}{\partial X_{mod,j}} \cdot M_{in,j}. \tag{11}$$

In tensor form:

$$\frac{\partial \mathcal{L}}{\partial X} = \frac{\partial \mathcal{L}}{\partial X_{mod}} \odot M_{in}. \tag{12}$$

Eq. equation 12 shows that the gradient signal is element-wise amplified by the mask $M_{in}$. Features in regions highlighted by the R-Model (where $M_{in} > 1$) receive stronger updates, encouraging the model to focus on these stable, task-relevant patterns from the outset.

**Gradient Analysis for Feature-level AGFM**  Here we analyze the influence on the intermediate feature map $F$. Using a similar derivation:

$$\frac{\partial \mathcal{L}}{\partial F} = \frac{\partial \mathcal{L}}{\partial F_{mod}} \odot M_{feat}. \tag{13}$$

Let $\theta_{feat}$ represent the encoder weights such that $F = f_{\theta_{feat}}(X)$. The gradient w.r.t. $\theta_{feat}$ is:

$$\frac{\partial \mathcal{L}}{\partial \theta_{feat}} = \left( \frac{\partial \mathcal{L}}{\partial F_{mod}} \odot M_{feat} \right) \frac{\partial F}{\partial \theta_{feat}}. \tag{14}$$

Eq. equation 14 demonstrates that the updates to the feature extractor are directly modulated by $M_{feat}$. By amplifying gradients for features identified as critical by the stable R-Model, AGFM enforces a "soft anchoring" effect, ensuring these representations are preserved and strengthened against interference from new tasks. This theoretical view confirms that PSAG actively shapes the optimization landscape to favor stability.

## A.2  Experimental Details and Analysis

### A.2.1  Hyperparameters and Training Details

**Base Configuration.**  Consistent with Shim et al. (2021); Gu et al. (2024), we employ a Reduced ResNet18 He et al. (2016) as the backbone. All methods are trained from scratch using SGD with a learning rate of

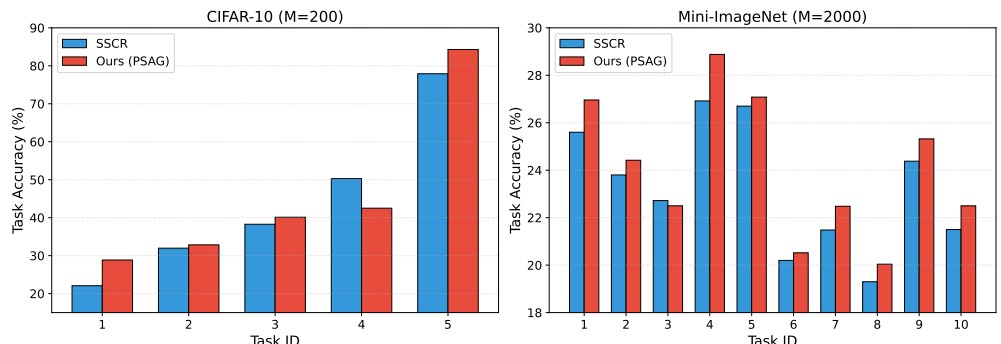

Figure 8: Task-wise accuracy comparison on CIFAR-10 ($M = 200$) and Mini-ImageNet ($M = 2000$). Bars represent accuracy on specific tasks after the full training sequence. PSAG shows consistent advantages across most tasks compared to SSCR, demonstrating effective preservation of old knowledge.

0.1. The batch size is set to 10 for the incoming stream and 64 for the memory buffer. We use memory buffer sizes of 200/500/1000 for CIFAR-10 Krizhevsky et al. (2009) and 1000/2000/5000 for CIFAR-100 Krizhevsky et al. (2009) and Mini-ImageNet Vinyals et al. (2016). The memory buffer sizes are set consistently with the compared methods to ensure a fair comparison.

**PSAG Hyperparameters.** We selected optimal hyperparameters based on a systematic grid search (detailed in Table 8). For CIFAR-10, we employ a balanced configuration with AGFM modulation weight $w = 2.0$ and IALR coefficient $\alpha = 2.0$, utilizing the standard EMA update schedule ($\eta_{init} = 0.1, \eta_{later} = 0.001$). For CIFAR-100, which presents a harder optimization landscape, we adopt more conservative guidance weights ($w = 0.5, \alpha = 0.5$) to prevent overfitting to the R-Model's initial representations. For Mini-ImageNet, we find that manifold projection plays a crucial role. Thus, we use $w = 0.5$ and $\alpha = 2.0$ (or $w = 1.0$ depending on buffer size) in conjunction with MCP to enforce geometric stability. The R-Model update frequency is fixed at $K = 50$ steps across all experiments.

### A.2.2 Additional Task-wise Analysis

To further validate the robustness of PSAG under different conditions, we present detailed task-wise accuracy breakdowns for two additional settings: a low-memory scenario (CIFAR-10, $M = 200$) and a complex dataset scenario (Mini-ImageNet, $M = 2000$).

Figure 8 compares the final accuracy of each task between PSAG and the strongest baseline (SSCR). In the challenging low-memory regime (CIFAR-10), PSAG demonstrates significant advantages on early tasks (e.g., Task 1 and Task 5), suggesting that attribution guidance is particularly effective when replay samples are scarce. Similarly, on Mini-ImageNet, PSAG maintains a consistent edge across the majority of the 10 tasks, further confirming its scalability.

### A.2.3 Extended Hyperparameter and Component Analysis

To provide a comprehensive view of the robustness and generalizability of the PSAG framework, we present detailed sensitivity analyses and ablation results.

**AGFM and Manifold Projection Analysis** Table 6 provides a granular breakdown of the performance of Attribution-Guided Feature Modulation (AGFM) across different modulation weights ($w \in [0.5, 3.0]$) and base methods (SCR vs. SSCR). We also explicitly compare the impact of enabling Manifold-Consistent Projection (MCP).

**Optimal Modulation Strength.** Across most configurations, a modulation weight of $w = 2.0$ consistently yields near-optimal performance. For instance, on CIFAR-10 with SSCR, the accuracy peaks at 57.39%

(Main Table) and remains robust around 57% for $w \in [1.5, 2.5]$. This suggests that AGFM provides a stable anchoring effect without requiring extremely precise parameter tuning.

**Effectiveness of MCP.** The results substantiate our claim regarding the dataset-dependent nature of geometric constraints. On the complex Mini-ImageNet dataset, enabling MCP consistently improves performance (e.g., SSCR+AGFM improves from 26.07% to 26.62% with MCP), confirming that Riemannian metric learning is beneficial for high-dimensional feature manifolds. Conversely, on the simpler CIFAR-10 dataset, MCP provides marginal or negligible gains, supporting our decision to prioritize AGFM and IALR for simpler tasks.

**Generalizability of IALR** Table 7 investigates the versatility of Importance-Aware Loss Reweighting (IALR) as a general-purpose plug-in module. We apply IALR to three distinct baselines: ER (random replay), SCR (contrastive replay), and SSCR (coreset-based contrastive replay).

**Generality across replay mechanisms.** IALR delivers consistent improvements across diverse base methods, validating that attribution-guided reweighting addresses a fundamental limitation, the blind equal treatment of samples, that persists across different architectures. Notably, the improvements are most significant on the vanilla ER baseline (e.g., +4.73% on Mini-ImageNet with $\alpha = 0.5$). Since ER lacks intrinsic feature regularization, it benefits immensely from the explicit sample prioritization provided by IALR.

**Complementary to sophisticated memory selection.** Even when applied to SSCR, which already employs sophisticated coreset selection to curate a high-quality memory buffer, IALR provides further gains (e.g., +1.31% on CIFAR-10). This indicates that memory selection (deciding *which* samples to store) and loss reweighting (deciding *how much* each sample contributes) represent orthogonal dimensions of optimization. IALR captures fine-grained, sample-level importance that is not fully encapsulated by coreset diversity metrics.

**Dataset-dependent dynamics.** The optimal weighting coefficient $\alpha$ varies by dataset complexity. CIFAR-10 generally benefits from stronger reweighting ($\alpha \in [1.0, 2.0]$), whereas the fine-grained classes of CIFAR-100 and Mini-ImageNet favor more conservative weights ($\alpha \in [0.1, 0.5]$). We also note a limitation that IALR does not improve SCR on CIFAR-100. We hypothesize that SCR's contrastive loss on this dataset may already implicitly emphasize hard negatives in a way that conflicts with IALR's explicit reweighting, warranting further investigation into these interactions.

**Joint Hyperparameter Space** Table 8 presents a comprehensive grid search over the joint hyperparameter space of AGFM ($w$) and IALR ($\alpha$) for the combined configuration.

**Synergy and Robustness.** The results reveal a broad sweet spot in the parameter landscape. For CIFAR-10, high performance ($> 56\%$) is achieved across a wide diagonal of configurations (e.g., $w = 2.0, \alpha = 2.0$ and $w = 3.0, \alpha = 0.5$). This indicates that the two mechanisms can trade off: stronger feature anchoring (high $w$) can compensate for weaker loss reweighting (low $\alpha$), and vice versa.

**Best Configurations.** Based on this grid search, we selected the balanced configuration of $w = 2.0$ and $\alpha = 2.0$ for CIFAR-10, and adjusted values for other datasets (e.g., lower $w, \alpha$ for CIFAR-100 to prevent overfitting) to report the final PSAG performance in the main text.

**Grid Search Analysis.** Table 8 shows that performance varies moderately across the tested hyperparameter grid. This relatively flat landscape suggests that PSAG is not overly sensitive to a narrow choice of hyperparameters, and maintains competitive performance across a reasonable range of configurations.

### A.2.4 Analysis of Forgetting

While Average Accuracy (ACC) measures overall performance, Average Forgetting (FGT) following the standard protocol Chaudhry et al. (2018) specifically quantifies the average performance degradation of each

Table 6: Detailed performance analysis of Attribution-Guided Feature Modulation (AGFM) with different modulation weights ($w$) and manifold projection settings. Results are reported as mean $\pm$ 95% confidence interval over 3 runs. **Bold** indicates the best weight for each configuration.

| Method | Dataset | AGFM Weight ($w$) | | | | | | Best |
|---|---|---|---|---|---|---|---|---|
| | | 0.5 | 1.0 | 1.5 | 2.0 | 2.5 | 3.0 | |
| SCR + MCP | CIFAR-10 | 55.02±2.92 | 55.95±2.56 | 55.34±1.97 | 55.26±1.60 | **56.40±3.35** | 55.72±2.09 | 56.40±3.35 |
| | CIFAR-100 | 27.51±1.45 | 27.61±1.33 | 26.77±2.49 | **28.15±1.01** | 27.54±1.12 | 26.97±1.98 | 28.15±1.01 |
| | Mini-ImageNet | 26.23±1.61 | 26.03±0.88 | 26.23±1.50 | 25.62±2.86 | 26.05±2.04 | **26.58±1.01** | 26.58±1.01 |
| SSCR + MCP | CIFAR-10 | 56.11±2.22 | 56.47±4.86 | **56.62±1.96** | 56.21±2.24 | 56.57±3.89 | 56.14±4.26 | 56.62±1.96 |
| | CIFAR-100 | 27.39±0.21 | 27.50±0.76 | 27.30±2.67 | 27.08±1.41 | **27.56±3.09** | 27.14±1.99 | 27.56±3.09 |
| | Mini-ImageNet | 26.52±2.22 | **26.62±1.19** | 25.96±2.12 | 25.73±2.36 | 25.68±3.07 | 25.36±1.32 | 26.62±1.19 |
| SCR (No MCP) | CIFAR-10 | 56.06±2.10 | 55.44±4.94 | **56.91±1.33** | 56.16±1.93 | 55.56±2.29 | 56.18±5.13 | 56.91±1.33 |
| | CIFAR-100 | 27.11±0.98 | **27.53±0.84** | 27.31±2.53 | 26.74±1.52 | 26.83±1.27 | 27.06±1.01 | 27.53±0.84 |
| | Mini-ImageNet | 25.56±1.92 | **26.12±0.48** | 25.32±1.10 | 25.42±1.28 | 25.71±0.41 | 25.93±1.12 | 26.12±0.48 |
| SSCR (No MCP) | CIFAR-10 | 56.82±6.82 | 57.07±1.84 | 56.80±5.27 | **57.54±3.97** | 56.69±1.85 | 56.97±1.20 | 57.54±3.97 |
| | CIFAR-100 | **27.34±0.76** | 27.08±1.69 | 26.62±2.23 | 27.11±1.29 | 27.27±2.45 | 27.09±0.75 | 27.34±0.76 |
| | Mini-ImageNet | 25.98±0.51 | 25.99±1.43 | 25.71±0.87 | **26.07±1.57** | 25.51±3.56 | 26.01±3.59 | 26.07±1.57 |

Table 7: Generalizability of Importance-Aware Loss Reweighting (IALR) across different base methods. We report accuracy (%) $\pm$ 95% confidence interval for varying loss weighting coefficients ($\alpha$).

| Base Method | Dataset | IALR Weight ($\alpha$) | | | | | Best |
|---|---|---|---|---|---|---|---|
| | | 0.1 | 0.5 | 1.0 | 2.0 | 5.0 | |
| ER | CIFAR-10 | 25.19±7.20 | 25.15±5.69 | **25.30±4.77** | 23.68±5.48 | 24.65±6.65 | 25.30±4.77 |
| | CIFAR-100 | **17.71±2.01** | 17.26±2.84 | 16.98±1.93 | 17.02±1.59 | 16.17±0.52 | 17.71±2.01 |
| | Mini-ImageNet | 18.60±2.73 | **18.90±2.91** | 18.86±2.39 | 16.66±2.65 | 16.25±5.65 | 18.90±2.91 |
| SCR | CIFAR-10 | 55.87±2.97 | **56.33±5.97** | 55.75±4.50 | 55.48±2.84 | 55.35±2.60 | 56.33±5.97 |
| | CIFAR-100 | **27.11±2.57** | 26.74±1.45 | 27.11±0.33 | 27.05±1.03 | 24.52±4.20 | 27.11±2.57 |
| | Mini-ImageNet | **25.78±1.35** | 25.77±2.37 | 25.73±1.63 | 25.72±1.50 | 25.08±2.53 | 25.78±1.35 |
| SSCR | CIFAR-10 | 55.32±2.63 | 56.41±2.37 | 56.89±2.77 | **57.29±3.13** | 57.05±5.00 | 57.29±3.13 |
| | CIFAR-100 | 27.21±1.65 | **27.53±1.37** | 26.33±1.32 | 27.18±1.63 | 25.52±1.98 | 27.53±1.37 |
| | Mini-ImageNet | **26.15±2.05** | 25.78±2.39 | 25.65±2.05 | 25.78±3.03 | 25.25±2.21 | 26.15±2.05 |

task from its peak accuracy to the end of training:

$$F_T = \frac{1}{T-1} \sum_{j=1}^{T-1} \max_{l \in \{j,...,T-1\}} (a_{l,j} - a_{T,j}).$$

Table 9 presents the FGT metric across all datasets and memory settings, where lower values indicate better stability.

**Competitive Stability.** PSAG demonstrates remarkable stability, consistently achieving lower forgetting rates than the strongest contrastive baseline (SSCR) in most configurations. For instance, on CIFAR-10 ($M = 500$), PSAG reduces forgetting to 25.38%, outperforming SSCR (27.20%). Similarly, on the challenging CIFAR-100 dataset ($M = 5000$), PSAG achieves a low forgetting rate of 6.41%, surpassing SSCR (7.28%).

**Stability-Plasticity Trade-off.** We note that certain methods, like SSIL, achieve lower forgetting rates in some settings (e.g., Mini-ImageNet). However, this often comes at the cost of plasticity, resulting in significantly lower overall accuracy compared to PSAG (as shown in Table 1). In contrast, PSAG strikes a superior balance: it maintains high plasticity to learn new tasks effectively while utilizing the R-Model's stable attribution signals to protect critical feature subspaces, thereby preventing catastrophic forgetting more effectively than other high-performance methods.

Table 8: Grid search results for combination (AGFM weight $w$ vs. IALR weight $\alpha$). We compare configurations with and without Manifold Projection (MCP). Results are mean $\pm$ 95% CI.

| Data | AGFM $(w)$ | IALR Weight ($\alpha$) - With Manifold (MCP) | | | | | IALR Weight ($\alpha$) - Without Manifold (No MCP) | | | | |
|---|---|---|---|---|---|---|---|---|---|---|---|
| | | 0.1 | 0.5 | 1.0 | 2.0 | 5.0 | 0.1 | 0.5 | 1.0 | 2.0 | 5.0 |
| CIFAR-10 | 0.5 | 56.25±4.26 | 55.67±3.56 | 55.30±1.96 | 55.69±3.00 | 55.57±5.28 | 55.69±4.16 | 56.41±3.62 | 55.94±3.25 | 56.31±3.52 | 56.46±3.19 |
| | 1.0 | 55.89±1.84 | 56.02±2.51 | 56.67±0.26 | 56.16±0.51 | 55.86±2.45 | 55.58±1.75 | 56.01±4.16 | 56.88±1.09 | 55.95±3.35 | 55.84±4.21 |
| | 1.5 | 55.31±1.30 | 56.25±2.80 | 56.46±2.24 | 55.70±1.67 | 55.89±2.39 | 55.60±1.80 | 55.58±1.04 | 55.49±2.71 | 55.58±3.88 | 56.52±2.17 |
| | 2.0 | 55.98±1.68 | 56.00±2.98 | 55.90±2.61 | 55.81±0.92 | 56.24±0.79 | 55.54±1.45 | 56.40±2.48 | 55.24±6.19 | 56.55±2.47 | 55.66±0.72 |
| | 2.5 | 54.82±1.18 | 55.89±0.68 | 55.72±2.50 | 55.50±2.19 | 56.86±4.04 | 56.59±2.65 | 56.17±2.31 | 56.27±2.25 | 54.35±1.59 | 55.97±3.45 |
| | 3.0 | 55.92±1.16 | 56.81±1.67 | 56.16±1.93 | 56.15±1.20 | 56.14±2.97 | 57.03±1.82 | 55.59±2.70 | 56.72±3.44 | 55.48±3.28 | 55.57±1.77 |
| CIFAR-100 | 0.5 | 27.57±0.33 | 28.50±0.37 | 27.78±1.55 | 27.47±0.59 | 27.64±1.40 | 26.26±2.83 | 27.21±1.03 | 27.23±1.44 | 27.45±1.22 | 27.10±1.53 |
| | 1.0 | 27.08±0.57 | 27.29±1.05 | 27.63±0.54 | 27.33±1.20 | 27.38±2.39 | 27.13±0.37 | 27.69±1.63 | 26.90±1.78 | 26.92±0.42 | 27.28±2.27 |
| | 1.5 | 27.09±1.62 | 28.06±1.67 | 27.45±1.69 | 27.41±1.70 | 28.38±1.50 | 26.64±2.70 | 26.82±1.40 | 26.96±0.48 | 26.48±1.07 | 27.00±1.72 |
| | 2.0 | 27.05±1.67 | 26.60±0.86 | 27.20±1.82 | 27.03±1.00 | 27.47±1.43 | 27.23±1.11 | 26.78±1.99 | 26.54±3.01 | 27.08±0.95 | 27.14±0.74 |
| | 2.5 | 27.40±0.76 | 27.33±2.34 | 27.61±0.54 | 26.88±1.85 | 27.60±1.06 | 26.95±1.41 | 26.14±1.33 | 26.64±1.14 | 27.11±0.54 | 27.21±1.32 |
| | 3.0 | 27.23±1.94 | 26.94±1.43 | 27.35±1.45 | 27.19±3.16 | 26.86±2.11 | 26.43±1.31 | 26.68±1.87 | 27.38±2.28 | 27.51±1.24 | 27.29±2.39 |
| Mini-Img | 0.5 | 26.45±0.67 | 25.90±1.83 | 26.28±1.37 | 26.60±0.93 | 25.95±1.72 | 25.50±1.85 | 25.77±0.50 | 25.73±0.90 | 25.62±2.05 | 25.43±1.75 |
| | 1.0 | 25.64±1.99 | 26.06±2.39 | 25.90±0.46 | 26.58±1.96 | 26.23±1.64 | 25.71±1.66 | 25.53±0.88 | 25.28±1.62 | 25.60±1.37 | 25.57±1.92 |
| | 1.5 | 26.34±1.80 | 25.90±2.47 | 25.82±1.63 | 26.23±1.44 | 25.77±2.07 | 25.42±2.35 | 25.58±0.70 | 25.48±0.36 | 25.32±1.47 | 25.35±2.23 |
| | 2.0 | 25.88±1.44 | 25.98±1.74 | 25.93±2.10 | 25.41±1.70 | 25.72±1.19 | 25.28±1.46 | 25.24±0.46 | 25.50±1.71 | 24.81±0.87 | 25.62±0.16 |
| | 2.5 | 25.97±0.41 | 25.79±1.80 | 26.40±1.92 | 25.69±2.03 | 25.53±0.98 | 25.66±0.94 | 25.69±2.99 | 25.47±0.65 | 25.26±0.26 | 25.65±1.11 |
| | 3.0 | 25.85±1.17 | 25.52±0.70 | 25.87±0.93 | 26.17±0.97 | 25.75±1.32 | 25.13±0.54 | 25.21±0.52 | 25.49±1.72 | 25.45±1.33 | 25.67±0.30 |

Table 9: Average Forgetting (FGT) comparison on CIFAR-10, CIFAR-100, and Mini-ImageNet across different memory buffer sizes $(M)$. Results are reported as mean $\pm$ 95% confidence interval over 5 runs.

| Methods | CIFAR-10 | | | CIFAR-100 | | | Mini-ImageNet | | |
|---|---|---|---|---|---|---|---|---|---|
| | $M = 200$ | $M = 500$ | $M = 1000$ | $M = 1000$ | $M = 2000$ | $M = 5000$ | $M = 1000$ | $M = 2000$ | $M = 5000$ |
| ER | 68.29±1.99 | 64.47±2.54 | 60.29±3.89 | 47.21±0.93 | 43.99±1.58 | 38.84±0.92 | 39.58±1.62 | 39.33±3.02 | 34.35±2.12 |
| MIR | 67.30±3.22 | 63.58±2.75 | 56.65±4.79 | 46.62±1.47 | 43.94±1.50 | 40.06±2.51 | 40.25±1.00 | 38.56±1.76 | 33.68±2.69 |
| ASER | 63.91±8.19 | 58.74±5.79 | 51.28±7.71 | 48.78±1.03 | 42.88±0.76 | 33.87±1.52 | 36.62±2.45 | 33.34±1.28 | 29.15±1.89 |
| SSIL | 21.85±3.49 | 17.37±2.48 | 15.83±3.40 | 15.36±2.45 | 13.45±1.71 | 10.11±2.10 | 13.52±0.56 | 13.15±1.33 | 9.43±2.31 |
| ACE | 29.16±4.34 | 26.53±5.40 | 24.60±4.42 | 16.45±1.60 | 14.36±1.52 | 10.59±0.89 | 19.92±2.56 | 13.15±1.33 | 11.84±2.40 |
| LAS | 65.17±5.37 | 53.31±4.87 | 40.36±6.72 | 38.07±2.04 | 29.87±3.05 | 21.27±2.45 | 30.04±1.61 | 23.67±3.25 | 19.82±3.31 |
| SCR | 39.64±4.65 | 26.27±1.25 | 18.72±1.11 | 12.81±0.89 | 8.65±1.02 | 6.89±1.17 | 11.06±0.76 | 8.70±1.62 | 7.08±1.05 |
| SSCR | 38.04±5.62 | 27.20±2.09 | 18.39±1.61 | 12.59±1.37 | 10.03±1.29 | 7.28±1.23 | 10.65±1.26 | 9.23±0.94 | 6.39±1.31 |
| PSAG | 38.01±2.10 | 25.38±2.40 | 18.39±1.52 | 12.65±0.96 | 9.04±0.80 | 6.41±0.90 | 10.49±0.78 | 8.46±1.47 | 6.48±1.58 |

### A.2.5 Implementation of Mechanism Analysis (PSAG-ER)

To conduct the fine-grained error and norm analysis presented in Section 4.5, we utilized a simplified variant of our framework, denoted as **PSAG-ER**. This variant applies the attribution guidance mechanism (AGFM) to the vanilla Experience Replay (ER) baseline.

**Dual-Level Modulation.** Unlike the main PSAG framework which focuses primarily on feature-level modulation for contrastive learning, PSAG-ER applies modulation at two distinct levels to explicitly visualize the impact of guidance on both input sensitivity and feature representation.

At the **Input-level (Pixel Space)**, we compute a spatial attention mask $M_{in} \in [0,1]^{H \times W}$ from the R-Model's attribution map $A$ and modulate the input image as $X' = X \odot (1 + \beta_{in} \cdot M_{in})$, thereby highlighting task-relevant pixels directly. Simultaneously, at the **Feature-level (Latent Space)**, consistent with the main framework, we resize the attribution map to match the feature dimensions to yield a mask $M_{feat}$, and modulate the intermediate features as $F' = F \odot (1 + \beta_{feat} \cdot M_{feat})$.

**Training Objective.** Since PSAG-ER is built on the standard ER baseline (without contrastive learning), it optimizes the standard Cross-Entropy loss using the modulated inputs and features. The complete training procedure is detailed in Algorithm 2. For the mechanism analysis experiments, we set $\beta_{in} = 0.05$ and $\beta_{feat} = 0.25$ based on the sensitivity study on CIFAR-10.

---

**Algorithm 2** PSAG-ER: Variant for Mechanism Verification

---

**Require:** Model $f_\theta$ (feature extractor $\psi$, classifier $h$), Data stream $\mathcal{S}$, Memory $\mathcal{M}$, Modulation strength $\beta_{in}, \beta_{feat}$

1: Initialize R-Model $\theta_{ref} \leftarrow \theta$
2: **for** Task $\mathcal{T}_i$ in $\mathcal{S}$ **do**
3:     **for** Mini-batch $(X_i, Y_i)$ in $\mathcal{T}_i$ **do**
4:         $(X_\mathcal{M}, Y_\mathcal{M}) \leftarrow \text{Sample}(\mathcal{M})$
5:         $X \leftarrow [X_i; X_\mathcal{M}], \quad Y \leftarrow [Y_i; Y_\mathcal{M}]$
6:         $A \leftarrow A(X, Y, f_{\theta_{ref}})$
7:         $M_{in} \leftarrow \text{Sigmoid}(\text{Normalize}(A))$               ▷ Generate soft pixel mask
8:         $X_{mod} \leftarrow X \odot (\mathbf{1} + \beta_{in} \cdot M_{in})$              ▷ Highlight critical pixels
9:         $F \leftarrow \psi(X_{mod})$                             ▷ Forward pass with modulated input
10:        $M_{feat} \leftarrow \text{Resize}(M_{in})$                  ▷ Align mask to feature dim
11:        $F_{mod} \leftarrow F \odot (\mathbf{1} + \beta_{feat} \cdot M_{feat})$          ▷ Modulate latent features
12:        $\mathcal{L} \leftarrow \text{CrossEntropy}(h(F_{mod}), Y)$
13:        Update $\theta$ using $\nabla_\theta \mathcal{L}$
14:        Update $\mathcal{M}$ with $(X_i, Y_i)$
15:        **if** $it \bmod K = 0$ **then**
16:           $\theta_{ref} \leftarrow (1 - \eta) \cdot \theta_{ref} + \eta \cdot \theta$
17:        **end if**
18:     **end for**
19: **end for**

---

Table 10: Average attribution norms of current and memory batches. Memory samples consistently exhibit higher attribution energy, validating their higher semantic density.

| Method | Current Batch Avg Norms | | | Memory Batch Avg Norms | | | Accuracy |
|---|---|---|---|---|---|---|---|
| | L1 | Inf | Fro | L1 | Inf | Fro | |
| ER | 490.99 | 1.44 | 13.24 | 279.15 | 0.81 | 7.38 | 36.42 |
| MIR | 317.22 | 0.98 | 8.44 | 346.09 | 1.09 | 9.21 | 40.16 |
| ASER | 283.47 | 0.91 | 7.64 | 263.53 | 1.01 | 7.53 | 34.63 |
| GSS | 249.13 | 0.82 | 6.79 | 223.89 | 0.72 | 6.05 | 31.47 |
| SSIL | 418.35 | 1.59 | 11.58 | 243.46 | 0.90 | 6.87 | 38.31 |

### A.2.6 Detailed Analysis of Attribution Dynamics

To further justify the design of PSAG, particularly the use of the R-Model, we conducted the in-depth analysis combining qualitative visualization and quantitative stability metrics.

**Norm Disparity with Memory and Current** We analyzed the magnitude of attribution maps using $\ell_1$, $\ell_\infty$, and Frobenius norms. Table 10 and Figure 9 reveal a significant disparity: attribution norms for memory samples are consistently higher than those for current stream samples. This phenomenon suggests that memory samples, having been filtered and replayed, carry denser semantic information that contributes more substantially to the model's decision boundary. This observation directly motivates our **IALR** (Importance-Aware Loss Reweighting) mechanism, which explicitly leverages this norm disparity to prioritize high-information samples, thereby amplifying the stabilizing effect of the replay buffer.

**Attribution Quality Degradation** Figure 10 visualizes the evolution of attribution maps for samples from Task 1 as the model learns subsequent tasks.

**The Problem of Drift.** In standard training, we observe a clear degradation in attribution quality: maps for early tasks (e.g., Task 1) start with focused attention on object shapes but gradually become scattered and noisy as the model overfits to later tasks (Tasks 4-5). This shift from semantic shapes to spurious texture correlations is a hallmark of catastrophic forgetting.

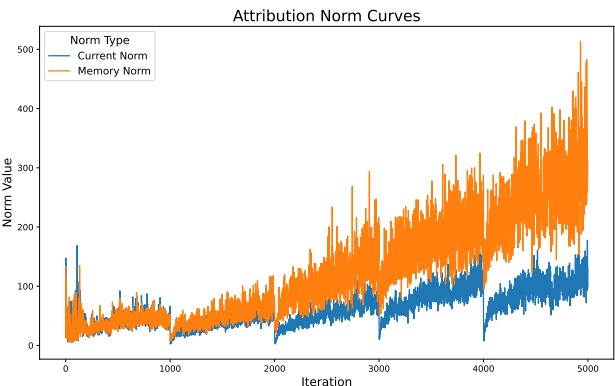

Figure 9: Evolution of attribution norms for current and memory samples in ER. The divergence in norms highlights the distinct roles of new and replayed data.

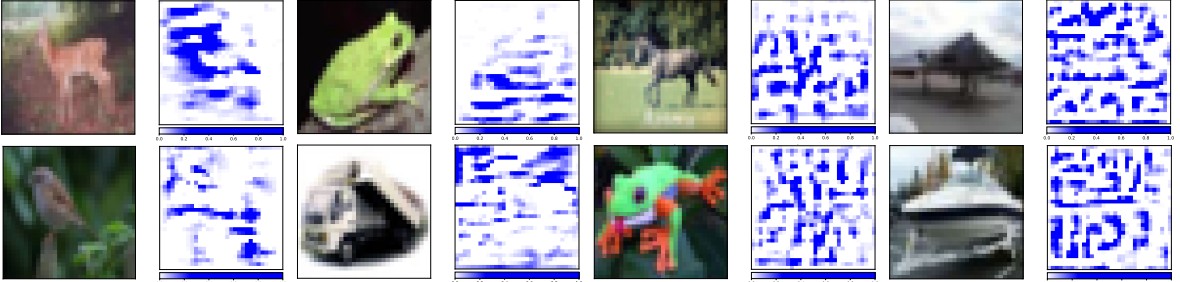

Figure 10: Visualization of attribution evolution. As training progresses (left to right), the model's attributions on new tasks become scattered, indicating forgetting across the current (upper row) and memory samples (lower row). PSAG's explicit guidance is needed to fully preserve semantic attention.

**The PSAG Solution.** This degradation justifies the necessity of our **R-Model**. By maintaining a slowly updating reference, the R-Model preserves the focused attention patterns of earlier states. PSAG then distills these stable patterns back into the current learner via AGFM, effectively counteracting representational drift and maintaining focus on discriminative features.

**Quantitative Evaluation of Stability** In addition to the qualitative visualizations in Figure 10, we quantify attribution stability by tracking the cosine similarity of attribution maps for a fixed reference batch of 64 hold-out images across consecutive task transitions.

Table 11 reports the results on Split CIFAR-10. The R-Model maintains consistently high attribution similarity, indicating that the EMA mechanism filters out high-frequency fluctuations from the plastic learner. In contrast, the current model exhibits larger attribution drift during task transitions. This supports the use of the R-Model as a stable attribution source for guidance.

Table 11: Quantitative evaluation of attribution stability measured by mean cosine similarity $\pm$ standard deviation on Split CIFAR-10.

| Memory Size ($M$) | Current Model Stability | R-Model Stability | Stability Improv. |
|---|---|---|---|
| $M = 200$ | $0.9044 \pm 0.0679$ | $0.9999 \pm 0.0001$ | $+10.55\%$ |
| $M = 500$ | $0.9538 \pm 0.0267$ | $0.9999 \pm 0.0001$ | $+4.84\%$ |
| $M = 1000$ | $0.8664 \pm 0.1034$ | $0.9998 \pm 0.0001$ | $+15.41\%$ |
| **Average** | 0.9082 | 0.9999 | $+10.27\%$ |

### A.2.7 Robustness and Statistical Analysis

**Statistical Significance and Effect Size.** We compare PSAG against SSCR over 5 runs across all 9 experimental settings using independent two-sample $t$-tests and Cohen's $d$. Table 12 reports the corresponding $p$-values and effect sizes. Since the number of runs is limited ($N = 5$), some settings show marginal $p$-values despite consistent accuracy improvements. We therefore report Cohen's $d$ as a complementary measure of effect size. PSAG achieves a large effect size ($d > 0.8$) in 8 out of 9 settings, with the remaining setting showing a medium effect. These results suggest that the observed improvements are consistent and practically meaningful across datasets and memory budgets.

Table 12: Statistical analysis comparing PSAG against SSCR across 9 settings over 5 runs. Cohen's $d > 0.8$ is typically interpreted as a large effect size.

| Dataset | Memory | P-value | Cohen's $d$ |
|---|---|---|---|
| CIFAR-100 | $M = 5000$ | **0.0178** ($*$) | 1.879 |
| Mini-ImageNet | $M = 1000$ | **0.0318** ($*$) | 1.642 |
| CIFAR-100 | $M = 1000$ | 0.0573 (n.s.) | 1.403 |
| CIFAR-100 | $M = 2000$ | 0.0890 (n.s.) | 1.224 |
| CIFAR-10 | $M = 500$ | 0.1559 (n.s.) | 0.991 |
| Mini-ImageNet | $M = 2000$ | 0.1543 (n.s.) | 0.995 |
| CIFAR-10 | $M = 1000$ | 0.1579 (n.s.) | 0.985 |
| Mini-ImageNet | $M = 5000$ | 0.1708 (n.s.) | 0.952 |
| CIFAR-10 | $M = 200$ | 0.2872 (n.s.) | 0.721 |

**Sensitivity to Attribution Methods** We evaluate PSAG with three attribution methods: Gradient w.r.t. Input, Integrated Gradients, and DeepLIFT. As shown in Table 13, the performance differences are relatively small across these choices. This suggests that PSAG can leverage different gradient-based attribution signals and is not tied to a single attribution implementation.

Table 13: Performance comparison with different attribution methods on Split CIFAR-10.

| Attribution Method | $M = 200$ | $M = 500$ | $M = 1000$ |
|---|---|---|---|
| Gradient (Vanilla) | 45.34 | 55.88 | 61.74 |
| Integrated Gradients | 45.78 | 54.82 | 61.57 |
| DeepLIFT (Default) | 46.21 | 55.53 | 62.02 |

### A.2.8 Mechanism Interactions and Limitations

**Analysis of Component Interactions.** PSAG integrates three attribution-guided mechanisms that operate at different levels of the learning process:

- **AGFM (Feature-Level):** Applies a spatial constraint on intermediate feature representations to encourage the encoder to retain important activation patterns.

- **IALR (Loss-Level):** Applies a scalar weight to the objective function, prioritizing samples with higher attribution energy.

- **MCP (Metric-Level):** Reshapes the geometry of the embedding space by incorporating attribution-induced importance into the contrastive metric.

Since these mechanisms operate at the feature, loss, and metric levels, they provide complementary forms of guidance. As shown in the ablation study, their empirical contributions are dataset-dependent rather than uniformly additive.

**Limitations and Failure Modes**   We identify two limitations related to the stability-plasticity trade-off under strong distribution shifts:

- **Delayed Adaptation:** Under abrupt concept drift, the R-Model may initially delay adaptation to new distributions. This reflects the inherent trade-off between maintaining stability and rapidly adapting to new information.

- **Risk of Representation Anchoring:** The R-Model may over-anchor representations to early tasks. In scenarios with strong concept drift, this could suppress newly emerging features. Future work could explore adaptive update rates or drift-aware schedules to mitigate this risk.

### A.2.9   Extension to NLP Domain

As a preliminary evaluation beyond vision tasks, we test PSAG on the 20Newsgroups text classification dataset using TF-IDF features and an MLP backbone. PSAG achieves 68.42% average accuracy, outperforming the Experience Replay (ER) baseline of 65.18% by 3.24 percentage points. This result suggests that stabilized attribution guidance may extend beyond image classification, although broader evaluations on more complex non-vision domains remain future work.

### A.2.10   R-Model Scheduling Ablation

We analyze the impact of the R-Model update rate ($\eta$) and update frequency ($K$). As shown in Table 14, the stable-phase update rate $\eta = 0.001$ gives the best accuracy among the tested values. We also find that updating every $K = 50$ steps outperforms more frequent ($K = 10$) or sparser ($K = 100$) updates. This supports the use of a slow-update reference model to balance stability and adaptation.

Table 14: Impact of R-Model update rate $\eta$ on accuracy (Split CIFAR-10, $M = 500$).

| Update Rate ($\eta$) | 0.1 | 0.01 | 0.001 | 0.0001 |
|---|---|---|---|---|
| **Accuracy (%)** | 54.82 | 56.45 | 57.39 | 56.15 |

### A.2.11   Detailed Runtime Comparisons

We profile the training cost of PSAG on CIFAR-10. As shown in Table 15, attribution computation adds approximately 2.75 ms per iteration, while the EMA update incurs a relatively small additional cost. The R-Model increases training-time computation, but it is not required during inference.

Table 15: Training time profiling per iteration on CIFAR-10.

| Component | Time (ms/iter) | Percentage |
|---|---|---|
| Backbone Forward/Backward | 4.90 | 47.2% |
| **Attribution Calculation** | **2.75** | **26.5%** |
| EMA Update | 0.91 | 8.8% |
| Other | 1.82 | 17.5% |
| **Total** | 10.38 | 100% |

Table 16 reports the runtime profiling on Mini-ImageNet. Despite the larger image resolution ($84 \times 84$) compared to CIFAR-10 ($32 \times 32$), attribution computation remains a moderate portion of the total training cost.

Table 16: Runtime profiling on Mini-ImageNet.

| Component | Time (ms/iter) | Percentage |
|---|---|---|
| Backbone Forward | 7.20 | 19.4% |
| **Attribution Calculation** | **7.41** | **19.9%** |
| EMA Update | 0.58 | 1.6% |
| Backward | 21.98 | 59.1% |
| **Total** | 37.16 | 100% |

### A.2.12 Attribution Source Analysis: Current Model vs. R-Model

To examine the effect of attribution source, we replace the R-Model attributions with attributions generated directly from the current model while keeping the remaining guidance mechanisms unchanged. As shown in Table 17, using current-model attributions reduces performance from 57.39% to 54.92% on Split CIFAR-10 with $M = 500$. This suggests that attributions from the rapidly changing learner can be noisy and that the R-Model provides a more reliable source of guidance.

Table 17: Ablation study on the source of attribution maps on Split CIFAR-10 with $M = 500$.

| Method | Attribution Source | Accuracy (%) | Drop |
|---|---|---|---|
| **PSAG (Ours)** | R-Model (Stable) | 57.39 | - |
| Ablation | Current Model (Noisy) | 54.92 | $\downarrow 2.47\%$ |

