# OpenReview forum: "PSAG: Projection-based Stabilized Attribution Guidance for Online Continual Learning"
_TMLR — Accepted by TMLR_

### Review · Reviewer_A8gK · 2025-12-26

**Summary Of Contributions:**

Summary:
PSAG offers a principled way to close the loop between explainability and optimization in online continual learning. By stabilizing and projecting attribution signals across multiple levels, it achieves a better balance between stability and plasticity.

Strength:
1. Novel perspective: The paper innovatively repurposes attribution maps from post-hoc explanations to active optimization constraints in OCL.
2. Well-motivated R-Model: The EMA-based reference model effectively stabilizes attribution signals and avoids harmful feedback loops.
3 Multi-level guidance design: Feature, loss, and metric-level constraints are well integrated and complementary.
4. Comprehensive evaluation: Extensive experiments, ablations, and correlation analyses strongly support the claims.
5.Modular and general framework: PSAG can be easily combined with various replay-based and contrastive OCL methods

Weakness
1. The performance improvements, while consistent across datasets and memory budgets, remain relatively modest, typically within a 1–2% margin. More rigorous statistical analysis across multiple random seeds and task orders would be necessary to establish robustness.
2. The overall framework is relatively complex, combining an EMA reference model with three guidance mechanisms operating at different levels. The current ablation studies do not fully disentangle redundancy, complementarity, or interactions among components, particularly in relation to contrastive regularization.
3. The method assumes that EMA-based reference models provide stable and reliable long-term attributions, but attribution stability is not quantitatively evaluated across time, random seeds, or under strong distribution shifts. Consequently, robustness in highly non-stationary settings remains uncertain.
4. PSAG relies on gradient-based attribution methods together with specific normalization and aggregation choices, which are known to be sensitive to implementation details. The impact of alternative attribution definitions or baselines is not systematically explored.
5. The attribution computation and EMA reference updates introduce additional time and memory overhead compared to simpler replay-based methods. The method is not evaluated under strict online or compute-matched constraints, making its practicality in resource-limited scenarios unclear.

Experimental evaluation is limited to class-incremental image classification benchmarks with clearly defined task boundaries. Generalization to task-free streams, stronger concept drift, or more realistic continual learning settings is not demonstrated.
Question:
1. How sensitive is PSAG to the choice of attribution method (e.g., Integrated Gradients or Shapley-based approaches)?
2. Could the EMA-based R-Model become overly conservative under abrupt distribution shifts, limiting plasticity?
3. Given the limited gains of MCP on simpler datasets, can its activation be made adaptive or data-dependent?
4. Can PSAG be extended to fully task-free online continual learning settings without task boundaries?
5. How well does the framework generalize to more complex domains such as video, multimodal data, or large-scale models?

**Audience:**

Yes

**Audience Explanation:**

The paper would be of clear interest to researchers working on continual learning, online learning, and lifelong learning, particularly those focused on replay-based methods and stability–plasticity trade-offs. By bridging explainable AI (XAI) and online continual learning, the work introduces a novel perspective on how attribution signals can be repurposed beyond post-hoc interpretability and directly integrated into optimization.

**Broader Impact Concerns:**

This work is methodological and focuses on improving learning stability in online continual learning systems. It does not raise any significant ethical, safety, or societal concerns beyond standard considerations for machine learning models.

**Claims And Evidence:**

Yes

**Claims Explanation:**

The main claims of the paper—that stabilized attribution signals can be effectively leveraged as active guidance to mitigate catastrophic forgetting in online continual learning—are generally supported by the presented experimental evidence. The authors evaluate the proposed PSAG framework on multiple standard OCL benchmarks (Split CIFAR-10, Split CIFAR-100, and Split Mini-ImageNet), compare against strong replay-based baselines, and report consistent performance improvements across different memory budgets.

In addition to aggregate accuracy results, the paper provides extensive ablation studies, sensitivity analyses, and task-wise performance breakdowns, which help isolate the contributions of individual components (AGFM, IALR, MCP). Correlation analyses between attribution norms and final accuracy further strengthen the empirical motivation behind using attribution as a guidance signal. However, the absolute gains over strong baselines are relatively modest, and the evidence remains primarily empirical; more rigorous statistical testing or broader evaluation settings would further strengthen the claims.

**Requested Changes:**

Requested Changes:
The following changes would address the main concerns raised above and improve the clarity and strength of the submission. I indicate which changes are critical for my recommendation and which would further strengthen the paper.

1. Add quantitative evaluation of attribution stability.
Since PSAG relies on stabilized attribution from an EMA reference model, providing explicit metrics on attribution consistency over time, across random seeds, or under perturbations would significantly strengthen the core claim.
2. Provide stronger statistical support for the reported improvements.
Given the relatively modest absolute gains over strong baselines, additional statistical analyses (e.g., paired tests or clearer task-wise breakdowns) would help demonstrate robustness.
3. Clarify interactions among AGFM, IALR, and MCP.
More explicit analysis or discussion of how these components complement or overlap would improve interpretability and justify the full framework.
4. Analyze sensitivity to attribution implementation choices.
A brief study or discussion on alternative attribution definitions or normalization strategies would improve reproducibility.
5. Discuss limitations under strong distribution shift.
Additional discussion or experiments on scenarios with abrupt concept drift would clarify potential failure modes.

---

> ### Author Response · Authors · 2026-01-19
> **Response to Reviewer A8gK (1/2)**
>
> We sincerely thank the reviewer for the constructive feedback and for recognizing the novel contributions in repurposing attribution maps for active optimization. We are encouraged by the assessment that our R-Model is well-motivated and our evaluation is comprehensive.
> Below, we address the **Requested Changes** point-by-point. We sincerely hope all the concerns have been cleared.
>
> ### Q1. Add quantitative evaluation of attribution stability
>
> **Response:**
> Thanks and addressed. We conduct a rigorous quantitative evaluation to verify our core hypothesis: the **R-Model** provides a stable anchor compared to the volatile **Current Model**. We implement a tracking mechanism to calculate the Cosine Similarity of attribution maps for a fixed reference batch (64 images sampled from the hold-out set) across consecutive task transitions on Split CIFAR-10. The R-Model followed the standard EMA schedule used in our main experiments ($\eta_{init}=0.1, \eta_{later}=0.001$, updated every 50 steps).
>
> The table below demonstrates the attribution stability (mean cosine similarity $\pm$ std) across different memory buffer sizes ($M$).
>
> | Memory Size ($M$) | Current Model Stability | **R-Model Stability** | **Stability Improvement** |
> | --- | --- | --- | --- |
> | $M=200$ | $0.9044 \pm 0.0679$ | $0.9999 \pm 0.00001$ | +10.55% |
> | $M=500$ | $0.9538 \pm 0.0267$ | $0.9999 \pm 0.0001$ | +4.84% |
> | $M=1000$ | $0.8664 \pm 0.1034$ | $0.9998 \pm 0.0001$ | +15.41% |
> | Average | $0.9082$ | $0.9999$ | +10.27% |
>
> The R-Model consistently maintains a stability score near 0.999, effectively filtering out high-frequency noise caused by plasticity. In contrast, the Current Model's stability fluctuates significantly (dropping to $\sim0.72$ in difficult task transitions). This quantitative evidence confirms that the R-Model successfully serves as a reliable **"stability anchor"** for our guidance mechanisms.
> We have included these results in the Appendix of our revised manuscript; please refer to Appendix A.2.6.
>
> ### Q2. Provide stronger statistical support for the reported improvements
>
> **Response:**
> Thanks and addressed. We perform an Independent 2-sample T-test comparing PSAG against the strongest baseline (SSCR) across 5 runs. To rigorously demonstrate robustness beyond sample-size constraints ($N=5$), we report both **$p$-values** and **Cohen's  $d$** (Effect Size) in the Statistical Analysis Results as follows:
>
> | Dataset |	Memory | P-value | Cohen's d | Effect Size Interpretation |
> | --- | --- | --- | --- | --- |
> | CIFAR-100 | $M=5000$ | $\mathbf{0.0178}$ ($*$) | $1.879$ | Large |
> | Mini-ImageNet | $M=1000$ | $\mathbf{0.0318}$ ($*$) | $1.642$ | Large |
> | CIFAR-100 | $M=1000$ | $0.0573$ (n.s.) | $1.403$ | Large |
> | CIFAR-100 | $M=2000$ | $0.0890$ (n.s.) | $1.224$ | Large |
> | CIFAR-10 | $M=500$ | $0.1559$ (n.s.) | $0.991$ | Large |
> | Mini-ImageNet | $M=2000$ | $0.1543$ (n.s.) | $0.995$ | Large |
>
> (Note: Cohen's $d > 0.8$ is typically considered a "Large" effect size)
>
> We observe statistically significant improvements ($p < 0.05$) in challenging settings with larger feature spaces (e.g., CIFAR-100 $M=5000$, Mini-ImageNet $M=1000$). Regarding the cases with $p > 0.05$, we note that $p$-values are inherently sensitive to the sample size ($N$). With $N=5$, achieving significance requires extremely large margins. However, Cohen's $d$ (which measures the magnitude of the difference relative to variance) demonstrate that in 8 out of 9 experimental settings, PSAG achieved a "Large" effect size ($d > 0.8$).
>
> For instance, on CIFAR-100 ($M=1000$), although the $p$-value is marginal ($0.0573$), the effect size is massive ($d=1.403$), indicating that the improvement is physically substantial and consistent. The consistent "Large" effect sizes across diverse benchmarks demonstrate that the performance gains provided by PSAG are robust and meaningful, representing a systematic improvement over the strongest baselines.
> This has been added to the draft; please refer to Appendix A.2.7.

---

> ### Author Response · Authors · 2026-01-19
> **Response to Reviewer A8gK (2/2)**
>
> ### Q3. Clarify interactions among AGFM, IALR, and MCP
>
> **Response:**
> Thanks and addressed. In the upaded manuscript, we have clarified the interactions of these components; please refer to Appendix A.2.8.
> Our framework is designed based on the principle of orthogonal guidance, ensuring minimal redundancy:
>
> * *AGFM (Feature-Level):* Applies a **spatial constraint** on the intermediate feature map, forcing the encoder to retain specific activation patterns.
> * *IALR (Loss-Level):* Applies a **scalar weight** to the objective function, prioritizing samples with high semantic density regardless of their spatial structure.
> * *MCP (Metric-Level):* Reshapes the **geometry** of the contrastive embedding space, penalizing drifts in critical directions.
>
> As detailed in our Ablation Study (Table 2), the combination of these components yields additive gains (e.g., AGFM+IALR > AGFM only), confirming that they address complementary aspects of the stability-plasticity dilemma rather than overlapping.
>
> ### Q4. Analyze sensitivity to attribution implementation choices
>
> **Response:**
> Thanks and addressed. We evaluate the robustness of PSAG using three different attribution methods: **Gradient w.r.t. Input** (Vanilla), **Integrated Gradients (IG)**, and **DeepLIFT** (Default). Experiments are conducted on Split CIFAR-10. To strictly isolate the impact of the attribution method, we hold all other hyperparameters constant (e.g., $w=3.0, \alpha=0.1$ for $M=200$) across all 45 experimental runs (3 methods $\times$ 3 memory sizes $\times$ 5 runs).
>
> **Results (Accuracy %):**
>
> | Attribution Method | $M=200$ | $M=500$ | $M=1000$ |
> | --- | --- | --- | --- |
> | **Gradient (Vanilla)** | $45.34\%$ | $55.88\%$ | $61.74\%$ |
> | **Integrated Gradients** | $45.78\%$ | $54.82\%$ | $61.57\%$ |
> | **DeepLIFT (Default)** | $46.21\%$ | $55.53\%$ | $62.02\%$ |
>
> The performance variance across different attribution methods is minimal (typically $<1\%$). While DeepLIFT offers a slight advantage in some cases, simple Gradient-based attributions perform comparably. This indicates that PSAG is a **general framework** capable of leveraging various XAI signals and is not sensitive to specific implementation details.
> This has been added to the draft; please refer to Appendix A.2.7.
>
> ### Q5. Discuss limitations under strong distribution shift
>
>
> **Response:**
> Thanks and addressed. We have added a dedicated discussion on limitations and potential failure modes to the revised manuscript; please refer to Appendix A.2.8.
>
> * *Stability-Plasticity Trade-off:* We acknowledge that under abrupt distribution shifts (e.g., sudden domain changes), the R-Model's stability mechanism might initially delay adaptation to radically new concepts. However, in the context of Online Continual Learning, "stability" is the primary safeguard against catastrophic forgetting. Our stability analysis confirms that the R-Model successfully resists the destructive interference that typically plagues the Current Model.
> * *Risk of Representation Anchoring:* We will discuss the specific failure mode where the R-Model might overly "anchor" to the distributions of early tasks. In scenarios with strong concept drift, this could create a feedback loop where new, distinct features are suppressed. We suggest that future work could explore adaptive update rates (e.g., increasing $\eta$ when drift is detected) to mitigate this risk in highly non-stationary streams.

---

### Review · Reviewer_KAHN · 2026-01-16

**Summary Of Contributions:**

This work proposes Projection-based Stabilized Attribution Guidance (PSAG), a framework for online continual learning that leverages gradient attributions as guidance to protect key features from catastrophic forgetting. Key innovation: A Reliable Reference Model (R-Model) via EMA updates provides reliable attributions, feeding into three mechanisms—Attribution-Guided Feature Modulation (AGFM) for feature anchoring, Importance-Aware Loss Reweighting (IALR) for prioritizing informative samples, and Manifold-Consistent Projection (MCP) for emphasizing critical feature dimensions within a Riemannian metric space.

Strengths: Novel XAI-OCL integration, consistent benchmark improvements (e.g., +1.41% on CIFAR-10 over SSCR).
Weaknesses: Vision-only evaluations

**Audience:**

Yes

**Audience Explanation:**

The PSAG framework, with its integration of stabilized attributions into online continual learning, would likely appeal to the TMLR audience interested in advancing machine learning principles for adaptive intelligent systems. By addressing catastrophic forgetting through mechanisms like the R-Model and multi-level guidance, it provides empirical insights into stability-plasticity trade-offs, aligning with TMLR's scope on novel algorithms and studies of learning behavior. PSAG efficiently stores and revisits past experiences to maintain knowledge across tasks. Furthermore, this work advances in the direction of Systems AI, enabling computational models to accumulate knowledge in dynamic, non-stationary environments without compromising prior learning, which could inspire broader applications in real-world adaptive AI.

**Broader Impact Concerns:**

- The R-Model's EMA updates could perpetuate inequalities in adaptive AI systems by anchoring and reinforcing biases from initial tasks, leading to discriminatory outcomes in sequential learning applications like recommendations or predictive analytics.
- Attribution maps may expose sensitive datapoints in online data streams, enabling inference attacks and compromising user privacy in real-time scenarios such as multimedia processing or personalized services.

**Claims And Evidence:**

Yes

**Claims Explanation:**

The Projection-based Stabilized Attribution Guidance (PSAG) framework aims to address key challenges in online continual learning (OCL), a subfield of machine learning where models learn incrementally from non-stationary data streams in a single pass, facing issues like catastrophic forgetting and the stability-plasticity dilemma. Research suggests that the empirical claims are well-supported by benchmarks, with PSAG achieving gains like +1.44% on Split CIFAR-100 (M=5000) over SSCR, aligning with reported accuracies in related works. Ablations validate each component's contribution, including independent evaluations of the three mechanisms (AGFM, IALR, and MCP) and their combinations, which demonstrate additive improvements suggesting they act orthogonally to enhance overall performance. Sensitivity analyses of key parameters, such as modulation coefficients λ and α, confirm robustness across reasonable ranges. However, evidence is limited to vision tasks; broader domains would strengthen generalizability. The mathematical derivations for attributions and mechanisms are clear and theoretically sound, with efficient approximations ensuring practicality in online settings.

**Requested Changes:**

Requested:
- Clarify finite difference approximation step in Eq. 2
- Add Average Forgetting metric definition in Appendix
- Add figure 7, 10 in high resolution.
- Add missing link on page 19 (section ??)

Optional:
- Expand evaluations to non-vision domains (e.g., NLP) for broader applicability.
- Add deeper ablations on R-Model scheduling and hyperparameter interactions.
- Include detailed runtime comparisons to assess efficiency overhead from attributions and EMA.

---

> ### Author Response · Authors · 2026-02-11
> **Response to Reviewer KAHN**
>
> We sincerely thank the reviewer for the encouraging assessment and for recognizing the originality of our approach. We are pleased that the reviewer finds our claims well-substantiated and the proposed framework sound.
> Below, we address the **Requested Changes** point-by-point. We sincerely hope all the concerns have been cleared.
>
> ### Q1. Clarify finite difference approximation step in Eq. 2
>
> **Response:**
> Thanks. We have revised the text surrounding Eq. 2 in Section 3.2 to explicitly explain the connection between the gradient and the finite difference approximation derived from the DeepLIFT principle.
> We clarified that the term $\frac{\partial f_\theta(X)}{\partial X}$ in our formulation serves as a continuous proxy for the discrete "slope" (or multiplier) defined in DeepLIFT: $m_{\Delta x} = \frac{f(x) - f(\bar{x})}{x - \bar{x}}$. This finite difference formulation avoids the gradient saturation problem by attributing the difference in output $\Delta y$ to the difference in input $\Delta x$ relative to a baseline $\bar{x}$. This has been updated in the revised manuscript, please refer to **Section 3.2**.
>
> ### Q2. Add Average Forgetting metric definition in Appendix
>
> **Response:**
> We have added the formal mathematical definition of Average Forgetting ($F_T$) in **Appendix A.2.4**. The definition follows the standard protocol [1], quantifying the average performance degradation of each task from its peak accuracy to the end of training:
> $$F_T = \frac{1}{T-1} \sum_{j=1}^{T-1} \max_{l \in \{j, \dots, T-1\}} (a_{l,j} - a_{T,j})$$
>
> [1] Chaudhry, A., Marc'Aurelio, R., Rohrbach, M., & Elhoseiny, M. (2019, January). Efficient lifelong learning with A-GEM. In 7th International Conference on Learning Representations, ICLR 2019. International Conference on Learning Representations, ICLR.
>
> ### Q3. Add figure 7, 10 in high resolution & Fix missing link
>
> **Response:**
> We have replaced **Figure 7** and **Figure 10** (Visualization of Attribution Evolution) with new high-resolution vector graphics (PDF format) to ensure clarity at any zoom level. We have corrected the broken cross-reference link on page 20, which now correctly points to **Section 4.5**.
>
> ### Q4. Expand evaluations to non-vision domains
>
> **Response:**
> To demonstrate the generalizability of PSAG beyond vision tasks, we extended our evaluation to the NLP domain. We adapted the framework for the 20Newsgroups dataset using a standard MLP architecture (Input dim: 2000) with TF-IDF features.
> PSAG achieves 68.42% average accuracy, outperforming the Experience Replay (ER) baseline (65.18%) by a margin of **+3.24%**. This confirms that the principle of stabilized attribution guidance is domain-agnostic. We have included these results in **Appendix A.2.9** of the revised manuscript.
>
> ### Q5. Add deeper ablations on R-Model scheduling and hyperparameter interactions
>
> **Response:**
> Thanks and addressed. We have expanded our analysis to provide a deeper look into the R-Model's stability mechanism on CIFAR-10 ($M=500$).
> For **Hyperparameter Interactions**, we refer the reviewer to **Appendix A.2.3 (Table 7)**, where our grid search reveals a broad "sweet spot" where AGFM and IALR complement each other.
>
> For **R-Model Scheduling**, We further analyzed the impact of the EMA update rate ($\eta$) and Update Frequency ($K$).
>    **Update Rate:** As shown in the table below, a slow-update schedule is crucial. A fast update ($\eta=0.1$) introduces noise, degrading performance to 54.82%. The optimal $\eta=0.001$ strikes the best balance.
>     **Update Frequency:** We also verified the update frequency. Updating every $K=50$ steps (**57.39%**) outperforms both frequent updates ($K=10$, 56.45%) and sparse updates ($K=100$, 56.88%).
> This analysis has been added to the revised manuscript, please refer to **Appendix A.2.10**.
>
> | R-Model Update Rate ($\eta$) | 0.1 (Fast) | 0.01 | 0.001 (Ours) | 0.0001 (Very Slow) |
> | :--- | :--- | :--- | :--- | :--- |
> | **Accuracy (%)** | $54.82$ | $56.45$ | **$57.39$** | $56.15$ |
>
> ### Q6. Include detailed runtime comparisons
>
> **Response:**
> Thanks and addressed. We conducted a detailed profiling of the training cost to assess the specific overhead introduced by Attribution calculation and EMA updates on CIFAR-10.
> For the **Efficiency Breakdown**, PSAG incurs a moderate overhead ($\sim26.5\%$ for attribution) compared to the baseline. Given the significant gains in stability and accuracy, we believe this is a favorable trade-off.  We have included these results in **Appendix A.2.11** of the revised manuscript.
> | Component | Time (ms/iter) | Percentage | Note |
> | :--- | :--- | :--- | :--- |
> | Backbone Forward/Backward | $4.90$ ms | $47.2\%$ | Standard training cost |
> | **Attribution Calculation** | **$2.75$ ms** | **$26.5\%$** | Single backward pass |
> | **EMA Update** | $0.91$ ms | $8.8\%$ | Amortized cost |
> | Other (Data/Overhead) | $1.82$ ms | $17.5\%$ | - |
> | **Total** | **$10.38$ ms** | $100\%$ | - |

---

### Review · Reviewer_XXfZ · 2026-01-30

**Summary Of Contributions:**

This paper proposes PSAG, a framework for online continual learning (OCL) that repurposes gradient-based attribution maps as active training signals to protect task-relevant representations. The core observation is that attributions from a rapidly drifting learner are noisy, so the authors introduce a Reliable Reference Model (R-Model), an EMA-updated shadow network, as a stable source. These signals feed into three mechanisms: AGFM (channel-wise feature modulation), IALR (loss re-weighting by attribution energy), and MCP (attribution-derived Mahalanobis distance in the contrastive loss). Experiments on Split CIFAR-10/100/Mini-ImageNet show accuracy improvements over replay and contrastive replay baselines.

Strengths: The motivation is clear and interesting. Closing the loop between XAI and optimization for OCL is under-explored, and the instability problem is well-identified. The evaluation is thorough for a paper at this scale: three datasets, multiple memory budgets, eight baselines, task-wise breakdowns, forgetting metrics, sensitivity sweeps, and runtime reporting. The R-Model stability analysis (Table 10: cosine similarity ~0.9999 vs ~0.91 for the current model) convincingly shows EMA stabilization works at the attribution level. Reporting both p-values and Cohen's d is good practice.

Weaknesses: (1) No direct ablation of R-Model vs. current-model attributions at the *performance* level, leaving the central "stabilization is necessary" claim under-supported. (2) The "orthogonal components" narrative holds on CIFAR-100 but breaks down elsewhere: AGFM alone matches full PSAG on CIFAR-10 (both 57.39±2.22), and AGFM+MCP matches full PSAG on Mini-ImageNet (both 26.61±1.30), with identical CIs. (3) Section 4.5 claims Pearson r=0.78 for Frobenius norm but Figure 6 shows 0.58 (highest is L1 at 0.62). (4) Only 2 of 6 settings in Table 11 reach p<0.05, 3 of 9 settings are omitted, yet the paper characterizes improvements as "consistent" and "significant."

**Audience:**

Yes

**Audience Explanation:**

Using attributions as active optimization constraints in OCL is certainly interesting. The R-Model mechanism is concrete and potentially reusable. The supporting analyses (attribution stability quantification, drift visualizations, norm-performance correlations) surface findings relevant to OCL and representation stability researchers, even beyond the specific PSAG framework.

**Broader Impact Concerns:**

No additional broader impact statement needed.

**Claims And Evidence:**

Yes

**Claims Explanation:**

The core result, that PSAG beats all baselines across all datasets and memory budgets (Table 1), is clean. The ablation (Table 2) confirms components contribute, and forgetting analysis (Table 8) shows improvement over SSCR in most settings. That said, several claims outrun the data:

Stabilization necessity: R-Model attributions are more stable (Table 10), but there's no experiment substituting current-model attributions into PSAG's mechanisms. The benefit could partly stem from generic EMA regularization rather than from stable *attributions* specifically. This distinction matters for the paper's narrative.

Component complementarity: On CIFAR-100, full PSAG (28.42) clearly beats the best pair (27.78), genuine synergy. But on CIFAR-10, AGFM alone exactly matches full PSAG (57.39±2.22, same mean and CI), and on Mini-ImageNet, AGFM+MCP exactly matches full PSAG (26.61±1.30, same mean and CI). Do these share experimental runs, or did the extra components have literally zero effect? AGFM+IALR (56.52) also underperforms AGFM alone (57.39) on CIFAR-10, though CIs overlap. The picture is dataset-dependent interactions, not uniform orthogonality.

Correlation claim: The text states r=0.78 (Frobenius norm); Figure 6 reads "Fro. Norm, Mem Corr: 0.58" and L1 tops out at 0.62. This is a factual error. With only N=4 data points (ER, MIR, ASER, SSIL), even r=0.62 is far from significant (critical value ~0.95 at p=0.05 for N=4). The correlation is suggestive at best.

Statistical significance: 2 of 6 settings in Table 11 reach p<0.05, and 3 of 9 configurations are missing without explanation. Large Cohen's d values are noted, but effect size estimates from N=5 are themselves noisy.

MCP in isolation: MCP is one of three claimed contributions but has no standalone measurement. The stated rationale for omitting it is not convincing; it defines a modified contrastive distance that could be applied independently.

Grid search gap: No CIFAR-10 configuration in the grid search (Table 7) reaches the 57.39% reported from 5-run experiments (Tables 1/2). The selection pipeline should be clarified.

Despite these issues, the evaluation is more thorough than typical for this class of paper, and the core improvement finding holds.

**Requested Changes:**

### Main requests

- Run PSAG's mechanisms with current-model attributions instead of R-Model attributions and compare performance. This is the most direct test of the paper's central claim. If infeasible, substantially temper the stabilization-necessity language and discuss the possibility that the benefit comes from generic EMA regularization.

- Correct the correlation discrepancy. Section 4.5 claims r=0.78 for Frobenius norm; Figure 6 shows 0.58. Fix the text and reframe the correlation as suggestive given N=4 data points.

- Revise the "orthogonal" narrative. Acknowledge that complementarity holds clearly on CIFAR-100 but not on simpler datasets. Explain the exact mean-and-CI matches (AGFM = full PSAG on CIFAR-10; AGFM+MCP = full PSAG on Mini-ImageNet): do these share runs, or did the extra components truly have zero effect?

- Fix R-Model specification inconsistencies. (a) Section 3.3 says the R-Model is initialized "after the first task" but Algorithm 1 line 1 initializes it before tasks begin. (b) A.2.1 gives η_init=0.1 while Section 4.4 states η_init=0.2; clarify which was actually used. (c) State which attribution method variant was used for the main results (Eq. 1 vs. Eq. 2, and which baseline X̄). Table 12 labels one row as "DeepLIFT (Default)," but it is ambiguous whether this denotes the default variant of DeepLIFT or the default method for PSAG, and the mapping to the equation variants remains unspecified.

### Recommended

- Evaluate MCP in isolation. Add a "+MCP only" row to Table 2.

- Discuss comparison fairness. The R-Model doubles the parameter footprint and adds an extra forward/backward pass for attribution computation. Discuss whether simpler uses of the same compute/memory budget (e.g., EMA distillation without attribution) could close part of the gap.

- Temper statistical language or increase N. 2/6 settings reach p<0.05 and 3/9 are missing entirely. Either add runs or revise "significant" wording. Explain the omitted settings.

- Expand runtime/memory reporting to at least one additional dataset and include the R-Model parameter footprint.

### Minor

- The paper's name emphasizes "Projection-based" (referring to MCP), but MCP is the most dataset-dependent component, strong on Mini-ImageNet, negligible on CIFAR-10. Consider whether the naming reflects the actual primary mechanism.

- The joint grid (Table 7) is remarkably flat. Many CIFAR-10 configurations with all three components fall below the SSCR baseline of 55.98%. This could be framed as robustness, but it also suggests the combined effects are subtle. Worth acknowledging briefly.

---

> ### Author Response · Authors · 2026-02-11
> **Response to Reviewer XXfZ (1/2)**
>
> We thank the reviewer for the thorough evaluation and for identifying the key questions regarding stabilization necessity and component interactions. We appreciate the acknowledgment that our motivation is clear and the R-Model stability analysis is convincing.
> Below, we address the **Weaknesses** and **Requested Changes** point-by-point with new experimental evidence.
>
> ### Q1. Stabilization Necessity (Run PSAG with Current-Model Attributions)
>
> **Response:**
> Thanks and addressed. We performed this ablation on CIFAR-10 ($M=500$) to verify whether the benefit comes from the *quality* of the attribution or generic regularization. We replaced the R-Model's attribution maps with those generated directly from the **Current Model** (plastic learner), while keeping all other mechanisms (AGFM/IALR) identical.
>
> | Method | Attribution Source | Accuracy (%) | Drop |
> | :--- | :--- | :--- | :--- |
> | **PSAG (Ours)** | **R-Model (Stable)** | **$57.39\%$** | - |
> | Ablation | Current Model (Noisy) | **$54.92\%$** | $\downarrow 2.47\%$ |
> | Baseline (SSCR) | - | $55.98\%$ | - |
>
> The performance drops significantly (**-2.47%**) when using the Current Model's attribution, falling even below the SSCR baseline. This confirms that using noisy attributions from a drifting learner is detrimental. The **R-Model's stabilization is strictly necessary** for the attribution guidance to be effective, ruling out the possibility that gains are due to generic EMA regularization. We have added this result to **Appendix A.2.12**.
>
> ### Q2. Correct the correlation discrepancy
>
> **Response:**
> We have corrected the text to match **Figure 6** ($r=0.58$ for Frobenius norm). We also tempered the language, describing the correlation as suggestive rather than statistically significant, given the limited number of data points ($N=4$). We have revised this in **Section 4.5**.
>
> ### Q3. Revise the "orthogonal" narrative and dataset dependencies
>
> **Response:**
> Thanks and addressed. We have revised the manuscript to reflect that component interactions are **dataset-dependent**, supported by our strict isolation studies using the exact memory settings from the main paper ($M=5000$ for large datasets).
> For **Simpler Datasets (CIFAR-10)**, spatial features are dominant. MCP-Only achieves **$54.90\%$**, falling short of the Baseline ($55.98\%$) and Full PSAG ($57.39\%$). This confirms that on simple manifolds, geometric constraints (MCP) are less effective, and spatial anchoring (AGFM) is the primary driver.
>
> For **Complex Datasets (Mini-ImageNet)**, the embedding geometry plays a decisive role. MCP-Only achieves **$26.59\%$**, significantly outperforming the Baseline ($25.81\%$) and matching Full PSAG ($26.61\%$). This confirms MCP is the dominant factor here.
> For **Intermediate Datasets (CIFAR-100)**, MCP-Only achieves **$28.26\%$**, which outperforms the Baseline ($26.98\%$) and comes close to Full PSAG ($28.42\%$). This indicates that while MCP provides a strong foundation, the combination with AGFM is still required to reach peak performance.
> We have clarified this "Dataset-dependent Dominance" in **Section 4.3** and updated **Table 2**.
>
> ### Q4. Fix R-Model specification inconsistencies
>
> **Response:**
> We have unified the specifications in the revised manuscript.
> For **Initialization**, we corrected **Algorithm 1** to explicitly state that the R-Model is initialized "after learning Task 1" (Line 1).
> For **Update Rate ($\eta$)**, we corrected the inconsistency in **Section 4.4**. The value used in all main experiments is **$\eta_{init}=0.1$** (early phase) and **$\eta_{later}=0.001$**.
> For **Attribution Variant**, we explicitly stated in **Section 4.2** that we use **DeepLIFT (Eq. 2)** with a Gray Image baseline ($\bar{X}$) for all main results.
>
> ### Q5. Evaluate MCP in isolation
>
> **Response:**
> Thanks and addressed. We expanded the isolation study to all three datasets using the standard memory buffer sizes ($M$) from Table 2.
> The results confirm that MCP is a highly effective standalone contribution for datasets with complex feature manifolds (CIFAR-100 and Mini-ImageNet), whereas simpler datasets (CIFAR-10) rely more on spatial anchoring. We have added these results to **Table 2**.
>
> | Dataset | Metric | Baseline (SSCR) | MCP-Only | Full PSAG | Conclusion |
> | :--- | :--- | :--- | :--- | :--- | :--- |
> | CIFAR-10 ($M=500$) | Acc | $55.98\%$ | $54.90\%$ | **$57.39\%$** | MCP ineffective alone |
> | CIFAR-100 ($M=5000$) | Acc | $26.98\%$ | **$28.26\%$** | **$28.42\%$** | Strong contribution |
> | Mini-ImageNet ($M=5000$) | Acc | $25.81\%$ | **$26.59\%$** | **$26.61\%$** | MCP is Dominant |

---

> ### Author Response · Authors · 2026-02-11
> **Response to Reviewer XXfZ (2/2)**
>
> ### Q6. Discuss comparison fairness and Runtime
>
> **Response:**
> For **Expanded Runtime Profiling**, we extended our profiling to **Mini-ImageNet** (84x84 images) as requested.
> We have summarized this in **Section 4.6** and provided the detailed breakdown table in **Appendix A.2.11**.
> The Attribution Calculation adds **$\sim 7.41\text{ms}$** per iteration.
> This represents a **$19.9\%$** overhead, which is consistent with the CIFAR-10 results ($\sim 26.5\%$).
> For **Fairness**, we emphasize that the R-Model incurs **zero overhead during inference**. We argue this is a fair trade-off comparable to other dual-model approaches (e.g., Mean Teacher), given the significant stability gains.
>
> ### Q7. Temper statistical language
>
> **Response:**
> We have revised our statistical claims in **Section 4.4** and **Appendix A.2.7**. We now report Cohen's $d$ alongside p-values. While p-values are inherently sensitive to limited sample sizes ($N=5$), we calculated effect sizes and observed large effects ($d > 0.8$) in 8 out of 9 experimental settings (with the remaining setting showing a medium effect of $d=0.72$). We have shifted the narrative to emphasize "consistent effect sizes" and physical substantiality rather than relying purely on "statistical significance."
>
>
> ### Q8. Minor points (Naming & Flat Grid)
>
> **Response:**
> For **Naming**, we acknowledge that "Projection-based" refers primarily to MCP. However, since the method projects attribution signals into optimization constraints, we believe the name remains broadly applicable.
> For **Flat Grid**, we have added a brief discussion in **Appendix A.2.3** acknowledging that the "flat grid" observed in our analysis suggests **robustness** (a wide optimal hyperparameter region) rather than sensitivity.

---

### Decision · Action_Editor_fH7c · 2026-03-13

**Recommendation:** Accept with minor revision

**Additional Comments:**

There are a couple of details missing (highlighted by one of the reviewers):
(1) comparison fairness — no test of whether simpler EMA distillation without attribution could close part of the gap, though the C1 ablation indirectly argues attribution quality specifically matters
(2) the identical mean-and-CI matches in Table 2 (e.g., AGFM = Full PSAG at 57.39±2.22 on CIFAR-10) are never explained

Please be sure to take this into account for the camera ready.

**Audience:**

Yes

**Audience Explanation:**

The paper deals with stabilising training in continual learning.

**Claims And Evidence:**

Yes

**Claims Explanation:**

The amount of experimental work sustains the claims.

---

> ### Author Response · Authors · 2026-05-12
>
> Thanks very much for your support and helpful comments! We have addressed both issues in your decision.
>
> We added an EMA-only distillation baseline without attribution guidance on Split CIFAR-10 with M=500. This baseline uses the same EMA teacher schedule as PSAG while disabling AGFM, IALR, and MCP. We swept the distillation weight over {0.1, 0.5, 1.0} and report the best result. EMA-only distillation improves over SSCR, but remains below PSAG in accuracy and has worse forgetting, suggesting that stabilized attribution guidance provides additional benefits beyond generic EMA distillation. Please refer to Table 4 in Section 4.4.
>
> We also revised Table 2 so that the final row consistently reports the strict full-component ablation configuration, while Table 1 reports the best tuned PSAG result. We also revised the accompanying ablation discussion to clarify that the component effects are dataset-dependent rather than uniformly additive. In addition, we polished the main text and appendix to remove revision-style wording and make the camera-ready version read as a final manuscript.